# Multimodal Policy Internalization for Conversational Agents

**Zhenhailong Wang**[1], **Jiateng Liu**[1], **Amin Fazel**[2], **Ritesh Sarkhel**[2], **Xing Fan**[2], **Xiang Li**[2],
**Chenlei Guo**[2], **Heng Ji**[2], **Ruhi Sarikaya**[2]
[1]University of Illinois Urbana-Champaign, [2]Amazon
wangz3@illinois.edu, jihj@amazon.com

## Abstract

Modern conversational agents such as ChatGPT and Alexa+ have become indispensable in everyday life. To handle diverse business requirements and enable agentic capabilities, these LLM-based systems often rely on predefined *policies*, which specify instructions such as model metadata, response styles, and tool-using rules. These policies, typically implemented as in-context prompts, are becoming increasingly complex and lengthy, posing challenges for models in faithfully following them. Moreover, they impose a large fixed computational cost regardless of the input query. As multimodal conversational agents emerge, complex policies that govern multimodal tasks and even involve visual instructions are becoming increasingly necessary, yet they have been rarely studied in previous work. In particular, prior work on prompt compression has focused solely on reducing the length of task templates and demonstrations, which require limited reasoning compared to policies. Meanwhile, related work on policy alignment has been limited to internalizing text-only safety instructions. To bridge this gap, we introduce **Multimodal Policy Internalization (MPI)**, a new task that aims to internalize reasoning-intensive multimodal policies into the parameters of a large multimodal model, enabling stronger policy-following behavior without requiring the policy to be included in-context during inference. MPI presents unique challenges from both data and algorithmic perspectives. We construct two new datasets that cover complex decision-making and tool-using tasks across both synthetic and real-world visual inputs. We investigate diverse internalization strategies and propose a novel three-stage training framework, **TriMPI**, which enables stronger guidance from the original policy during internalization. Specifically, we first introduce a continual pretraining stage before supervised finetuning, which directly injects policy knowledge into the model. We then propose **PolicyRollout**, a simple yet effective extension to GRPO-style RL algorithms, which enables more grounded exploration by augmenting the rollout space with policy-aware responses. We show significant improvements of TriMPI over strong baselines in end-to-end performance, generalization capability, and robustness to catastrophic forgetting. As the first work on multimodal policy internalization, we aim to build a strong foundation for future research by providing datasets, training recipes, and comprehensive evaluations.

## 1 Introduction

Conversational agents such as ChatGPT, Claude, and Alexa+ (OpenAI, 2025; Anthropic, 2025; Alexa AI, 2025) have become integral to daily life. To manage diverse business rules and enable agentic functionality, these LLM-based systems often rely on predefined *policies*, which are structured instructions that specify model metadata, response styles, tool-usage rules, and more. These policies, typically provided as in-context prompt prefixes, are becoming increasingly long and complex (estimated to range from ~1K to ~50K tokens[*]), imposing a substantial fixed computational cost regardless of the query size. In contrast, typical user queries usually range from 50 to 200 tokens (Clark et al., 2025), leading to a $20\times$ to $250\times$ higher input token cost from the policy prompts compared

---

[*]Exact numbers are not disclosed due to the proprietary nature of system prompts.

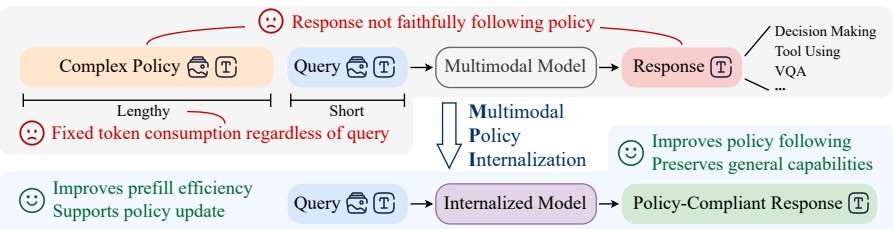

Figure 1: **Motivation of the proposed Multimodal Policy Internalization task.** The goal is to enhance the policy-following abilities of a large multimodal model without requiring the policy to be provided in-context during inference, thereby improving both performance and efficiency.

with the actual user queries. Moreover, as these policies expand and become more reasoning-intensive (e.g., requiring the model to follow rules distributed across different sections), models often struggle to adhere to them consistently (Yao et al., 2024; Qian et al., 2024). A natural research question arises: *can we internalize the knowledge of policies into the model parameters while also improving the model's policy-following abilities?*

Previous research on prompt compression (Li et al., 2024) has shown initial success in reducing token usage through hard prompting (Li et al., 2023; Jiang et al., 2023), soft prompting (Mu et al., 2023; Ge et al., 2023), or progressive fine-tuning (Zou et al., 2024). However, these approaches primarily focus on compressing templates and demonstration examples, which require minimal reasoning to adhere to the policy. More recent work (Guan et al., 2024) introduced deliberative alignment, which aims to internalize knowledge of more complex safety specifications and emphasizes improving policy-following performance beyond token reduction. Nonetheless, it remains limited to issues of model trustworthiness and has been explored only in text-only models. With the growing trend of multimodal conversational agents, policies are increasingly tied to multimodal tasks and may even include visual instructions such as demo images. Yet, no prior work has explored how to learn and internalize complex policies in multimodal models.

To address this gap, we propose a new task, **Multimodal Policy Internalization (MPI)**, which aims to train multimodal models that can generate policy-compliant responses without requiring the policy to be included in-context. Compared with prior work, the MPI task introduces several unique challenges: (1) The target policies focus on reasoning-intensive multimodal tasks, such as decision-making and tool usage for conversational agents. (2) There is a lack of existing datasets containing multimodal question-answer pairs that adhere to predefined policies. (3) There is a lack of established training paradigms for internalizing multimodal policies. For example, some text-only knowledge injection methods, such as continual pretraining (Ovadia et al., 2025; Maini et al., 2024), may not be directly applicable to MPI due to the presence of visual tokens.

In this work, we contribute from both data and algorithmic perspectives to advance research in multimodal policy internalization. We first construct two new datasets, ClevrPolicy and GTAPolicy, to support training and evaluation across different multimodal policy types. Specifically, ClevrPolicy focuses on internalizing complex decision-making policies that require multi-hop reasoning. Built upon the synthesized CLEVR dataset (Johnson et al., 2017), ClevrPolicy provides flexible control over policy complexity and dataset size, enabling in-depth investigation of the effectiveness of different algorithms. Curated from the GTA dataset (Wang et al., 2024a), GTAPolicy targets multimodal tool-usage instructions with real-world images and user queries. GTAPolicy emphasizes a low-data regime, where only limited question-answering data is available to demonstrate the expected behavior.

From the method perspective, a key challenge lies in how to align the response with the policy without assuming its existence during inference. Prior methods (Zou et al., 2024; Guan et al., 2024) focus solely on learning policy-following behavior, leaving it unclear how to better leverage the policy context itself as guidance during training. To address this, we introduce a novel three-stage training framework, **TriMPI**, which incorporates two key ideas: (1) warming up the model through continual pretraining directly on the policy context; and (2) a new RL algorithm, **PolicyRollout**, which enables additional conditioning on the policy context without introducing a gap between training and inference. Specifically, TriMPI consists of a visually-masked continual pretraining (VM-CPT) stage, a chain-of-thought supervised finetuning (CoT-SFT) stage, and a reinforcement learning (RL) stage with PolicyRollout. The VM-CPT stage enables language modeling directly on the multimodal policy, thereby explicitly injecting the entirety of the policy knowledge into the

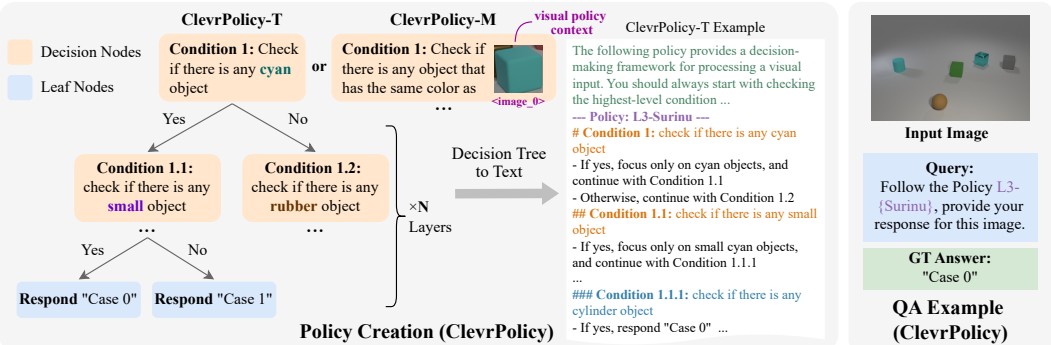

Figure 2: **ClevrPolicy dataset.** Left: Illustration of policy generation, where a decision tree is first generated and converted into natural language instructions (see Appendix C.1 for details on the decision node ontology, and Figures 18, 19 in the Appendix for full policy examples). Right: Example input-output pair corresponding to the policy. The policy is available only during training and not during inference.

model and facilitating reasoning in later stages. The RL stage is essential for internalizing reasoning-intensive policies, as it enables the model to learn from a broader range of policy-related responses through trail-and-error rather than memorization. PolicyRollout further enhances RL exploration by augmenting the rollout space with policy-grounded responses, serving as a simple yet effective extension to GRPO (Shao et al., 2024) and DAPO (Yu et al., 2025).

We demonstrate that TriMPI yields significant improvements in MPI, approaching 70.7% and 79.4% absolute gains over CoT SFT baseline and in-context setting, respectively. Beyond end-task performance, we also demonstrate enhanced generalization capability to policy updates and robustness against catastrophic forgetting. Further analysis shows that the improvements of TriMPI are consistent across different policy complexities and model sizes, with more pronounced gains observed on complex policies. To summarize, our main contributions are threefold: **(1) a new task** that targets the internalization of complex policies in the multimodal domain; **(2) two new datasets** that support both analytical and real-world training and evaluation; and **(3) a new training algorithm** that introduce an effective training paradigm with a customized RL algorithm for policy internalization.

## 2  PROBLEM FORMULATION

We formally define the proposed task, Multimodal Policy Internalization (MPI), as illustrated in Figure 1. Consider a multimodal conversational agentic task, where, given an input text query $Q$ and visual inputs $I$, the expected output is a textual response $A$, which can be either free-form natural language or structured code for tool calling. Additionally, the response behavior should follow the instructions predefined in a policy context $P$. Note that the policy may include both textual $P_T$ and visual $P_I$ components. Let $\mathcal{M}_\theta$ denote a multimodal model parameterized by $\theta$. The following illustrates the expected inputs and outputs before and after internalization:

$$A = \mathcal{M}_\theta(Q, I, P) \xrightarrow[\theta]{\text{Policy Internalization}} A = \mathcal{M}_\theta(Q, I) \qquad (1)$$

The goal is to embed the knowledge of the policy into the model parameters $\theta$, enabling better policy-compliant generation without requiring $P$ in-context during inference. MPI shares the same high-level motivation as deliberative alignment (Guan et al., 2024; Zhang et al., 2025), where we emphasize improving models' alignment with the policy beyond compressing the prompt. We do not consider training additional special embeddings as in soft prompting, since they are inherently tied to specific tasks (Li & Liang, 2021; Patel et al., 2025) and therefore limit the model's ability to maintain general reasoning capabilities and robustness (Fan et al., 2025; Bailey et al., 2023).

## 3  DATASET CREATION

### 3.1  CLEVRPOLICY DATASET

We first create ClevrPolicy, a new dataset focused on reasoning-intensive, visually dependent decision-making. ClevrPolicy is built upon images and scene graphs from the Clevr dataset (Johnson et al.,

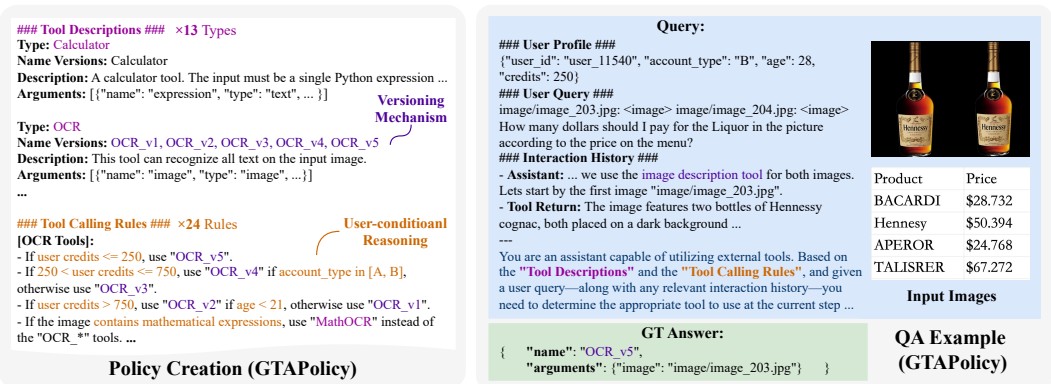

Figure 3: **GTAPolicy dataset.** Left: illustration of the policy, consisting of two major parts, tool description and tool calling rules (see Figure 20 in the Appendix for the full policy). Right: input and output example corresponding to the policy. The visual input can contain multiple images.

Table 1: **Zero-shot in-context performance** on the ClevrPolicy and GTAPolicy benchmarks. The metrics are reported as accuracy percentages (%) and are detailed in Appendix C.3. We observe a strong correlation between the number of decision-tree layers (N) and policy complexity. Introducing multimodal demonstrations into the policy (in ClevrPolicy-M) further increases the difficulty.

| Model | ClevrPolicy-T | | | ClevrPolicy-M | | | GTAPolicy | | |
|---|---|---|---|---|---|---|---|---|---|
| | N=2 | N=4 | N=6 | N=2 | N=4 | N=6 | Tool Acc | Arg Score | Overall |
| Qwen2.5-VL-3B | 36.60 | 10.85 | 4.80 | 33.05 | 8.05 | 4.55 | 18.87 | 15.44 | 17.15 |
| Qwen2.5-VL-7B | 72.00 | 32.20 | 13.15 | 51.20 | 13.90 | 5.65 | 23.58 | 19.44 | 21.51 |
| Claude-3.7-Sonnet | 97.65 | 92.60 | 85.35 | 95.00 | 82.75 | 77.63 | 42.45 | 37.13 | 39.79 |
| Claude-4-Sonnet | 98.10 | 96.70 | 90.10 | 93.40 | 78.55 | 77.76 | 60.38 | 51.53 | 55.96 |

2017), enabling fine-grained control over policy complexity and supporting comprehensive evaluation of different MPI algorithms. Here, policy complexity refers to how difficult it is to follow the instructions in the policy when it is provided in-context. To construct the policies, we generate binary decision trees in which decision nodes specify visual conditions and response nodes define actions, with tree depth determining complexity. Each decision tree is then converted into a structured natural language instruction as the final policy. We provide two variants of ClevrPolicy: **ClevrPolicy-T**, where policies are purely text-based, and **ClevrPolicy-M**, where policies include image demonstrations as part of the decision node conditions. **Figure 2** illustrates the policy creation process and shows an input-output example. Further details are provided in **Appendix C.1**.

## 3.2 GTAPOLICY DATASET

We further introduce **GTAPolicy**, which focuses on complex tool-using policies with real-world images and queries. We particularly consider a low-data regime, reflecting a common practical challenge in real-world settings where only limited QA pairs are available for training. The policy in GTAPolicy is constructed from the GTA dataset (Wang et al., 2024a), containing tool descriptions for 13 tools and 24 tool-calling rules with versioning and user-conditional mechanisms to simulate real-world business constraints. The policy can be automatically expanded when additional tools are provided. Training data reformulates multi-turn interactions into single-turn tool-calling tasks, where each instance includes visual inputs, a user profile, a query, an interaction history, and an instruction prompt, with the expected output being a JSON-formatted tool call specifying tool name and arguments. **Figure 3** illustrates the policy and an input-output example in GTAPolicy. Further details on the GTAPolicy dataset are provided in **Appendix C.2**. The statistics and evaluation metrics for both datasets are presented in **Appendix Table 7** and **Appendix C.3**, respectively.

## 3.3 ZERO-SHOT IN-CONTEXT RESULTS ON CLEVRPOLICY AND GTAPOLICY

We conduct an in-context evaluation on the proposed two benchmarks using off-the-shelf models, where the policy is directly inserted into the inference prompt. This evaluation provides a useful

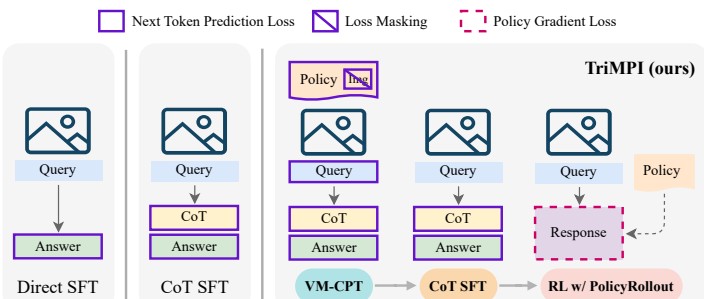

Figure 4: **Overview of different training algorithms** for multimodal policy internalization. The solid purple outlines indicate the parts where the next-token prediction loss is computed. On the right, we illustrate the proposed three-stage training strategy, TriMPI, which enables direct policy knowledge injection through the VM-CPT stage and policy-grounded reinforcement learning through PolicyRollout. The PolicyRollout algorithm is detailed in §4.3 and illustrated in Figure 5.

reference for assessing the complexity of the policies. The in-context performance also provides guidance on which methods to rely on for generating chain of thought (CoT) data for MPI. As shown in **Table 1**, on ClevrPolicy, performance decreases as the layer number increases. Even the strongest model, such as Claude-4 (Anthropic, 2025), begins to struggle at N=6 particularly on ClevrPolicy-M. GTAPolicy also poses significant challenges for all models, with the best tool accuracy only at 60%.

# 4 MULTIMODAL POLICY INTERNALIZATION (MPI) ALGORITHMS

## 4.1 BASELINES

Inspired by previous work on prompt compression (Zou et al., 2024) and deliberative alignment (Guan et al., 2024) in text domains, we consider the following baseline methods for MPI.

**Direct SFT.** We directly train on the non-CoT data using supervised finetuning (SFT), aiming to learn a mapping from the input to the answer without performing intermediate reasoning.

**CoT SFT.** We first obtain chain-of-thought (CoT) data in which the model explicitly reasons over the rules in the policy before producing the final answer, and then perform SFT on this CoT data. To generate CoT data for ClevrPolicy and GTAPolicy, we adopt different approaches depending on task complexity and data availability. Further details are provided in **Appendix D**.

## 4.2 TRIMPI: A THREE-STAGE TRAINING STRATEGY FOR MPI

We observe a key limitation in the baseline methods: they focus solely on learning the expected behavior without direct access to the original policy. This limitation becomes more evident for complex policies, where directly learning output patterns is increasingly difficult. This raises the following question: *can we better augment the learning process with the policy, without introducing a gap between training and inference?* Empirically, we find that simply inserting the policy into the prompt during training but removing it during inference results in near-random performance. To address this challenge, we propose **TriMPI**, a three-stage training framework that warms up the model via continual pretraining on the original policy and introduces **PolicyRollout**, a new RL algorithm that enables more policy-aware exploration (detailed in §4.3).

TriMPI consists of three stages: (1) Visually-Masked Continual Pretraining (**VM-CPT**); (2) Supervised Finetuning with Chain-of-thought (**CoT SFT**); (3) Reinforcement learning (**RL**). The CoT SFT stage is identical to the baseline described in §4.1. We present the details on the VM-CPT and RL stage next. An illustration of the baselines and TriMPI stages is presented in **Figure 4**.

**VM-CPT Stage.** This stage aims to inject policy knowledge directly into the model parameters before performing SFT. Specifically, we construct a new variant of the CoT dataset $D$, which contains input-output sequences $x = (P_T, P_I, I, Q, C, A)$ defined as the concatenation of tokens from the policy $P = (P_T, P_I)$, the visual inputs $I$, the text query $Q$, the CoT reasoning $C$, and the final answer $A$. We then compute the next-token prediction loss over all tokens except the visual tokens.

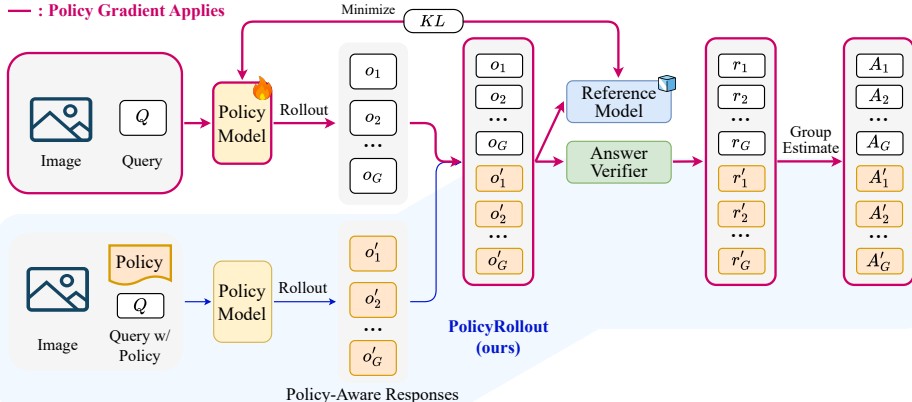

Figure 5: **Illustration of the PolicyRollout algorithm** (applied to GRPO as an example). During the rollout phase, we additionally construct a set of input instances with the policy included in-context. These policy-aware responses are added to the rollout space as if they were generated from the original inputs without the policy in-context. The advantage and policy gradient are then computed on the combined rollouts, indicated by the thick red outlines. PolicyRollout enables more policy-aware exploration without introducing a gap between training and inference, leading to significant improvements in MPI, especially on complex policies.

$$\mathcal{L}(\theta) = -\mathbb{E}_{x \sim \mathcal{D}} \left[ \frac{1}{\sum_{t=1}^{T} m_t} \sum_{t=1}^{T} m_t \log p_\theta(x_t \mid x_{<t}) \right], \quad m_t = \mathbf{1}[x_t \notin P_I \cup I]. \quad (2)$$

The visual masking $m_t$ enables us to adopt CPT (Ovadia et al., 2025; Maini et al., 2024) in this multimodal domain, where continuous visual tokens may appear in both the input $I$ and the policy $P_I$, and it has shown empirical success despite its simplicity.

**RL Stage.** As the target policy becomes more complex and reasoning-intensive, SFT baselines face the challenge of insufficient coverage of policy-related behaviors, particularly in low-data regimes. Inspired by recent advancements in reinforcement learning with verifiable rewards (RLVR), which can effectively learn from negative samples and exploration, we investigate its effectiveness for the MPI task. We first adopt GRPO (Shao et al., 2024) and DAPO (Yu et al., 2025), following a standard response format and reward design. Specifically, we ask the model to generate a thinking block enclosed within `<think></think>` and an answer block enclosed within `\boxed{}`, on which we compute both a format reward and an accuracy reward (detailed in Appendix C.3).

From our initial experiments, we observe that although GRPO and DAPO yield substantial empirical gains, exploration in the RL stage remains insufficiently grounded in the policy. This limitation becomes more pronounced with complex policies, where ungrounded exploration rarely produces positive rewards. Simply adding the policy in-context during training is ineffective, since the model lacks access to the policy at inference time, introducing a misalignment between training and inference. In the next section, we present our solution through a modified rollout phase.

## 4.3 POLICYROLLOUT (PoRo): POLICY-AWARE REINFORCEMENT LEARNING

To further improve the effectiveness of the RL stage for MPI, we introduce **PolicyRollout (PoRo)**, a simple yet effective extension to GRPO-style algorithms that **augments the rollout space with policy-aware responses**. Specifically, during the rollout stage, for each sampled instance, we construct a variant by inserting the policy in-context. We then allow the current policy model to generate an additional set of responses conditioned on the query $Q$, the image $I$, and the policy $P$. These policy-aware responses are concatenated with the no-policy responses to form the rollout space, after which we proceed with group-based advantage estimation. An illustration is provided in Figure 5, and the PolicyRollout objective (applied to GRPO as an example) is written as follows:

Table 2: **Main results and ablations** on multimodal policy internalization performance. By default, we use Qwen2.5-VL-7B as the base model. PoRo refers to PolicyRollout. The metrics are reported as percentages (%) and are detailed in Appendix C.3. We observe significant improvements of TriMPI over in-context and SFT baselines. Comprehensive ablations demonstrate the importance of each stage and the effectiveness of PolicyRollout. The RL steps indicate the actual update steps (for the three datasets, respectively, separated by "|"). Early stopping (marked with "*") may occur in DAPO (Yu et al., 2025) runs due to its dynamic sampling strategy. Notably, on DAPO, TriMPI achieves competitive or stronger performance while using fewer steps.

| Method | Stages | | | RL Steps | ClevrPolicy-T Acc (N=6) | ClevrPolicy-M Acc (N=6) | GTAPolicy | | |
| | CPT | SFT | RL | | | | Tool Acc | Arg Score | Overall |
|---|---|---|---|---|---|---|---|---|---|
| *Zero-Shot with Policy In-Context* | | | | | | | | | |
| In-Context | | | | − | 13.15 | 5.65 | 23.58 | 19.44 | 21.51 |
| *SFT Baselines for MPI* | | | | | | | | | |
| Direct SFT | | ✓ | | − | 15.15 | 14.55 | 44.34 | 37.16 | 40.75 |
| CoT SFT | | ✓ | | − | 17.80 | 14.30 | 57.55 | 51.45 | 54.50 |
| *TriMPI Ablation without RL Stage* | | | | | | | | | |
| VM-CPT + CoT SFT | ✓ | ✓ | | − | 22.75 | 27.05 | 69.81 | 61.13 | 65.47 |
| *TriMPI Ablation without VM-CPT Stage* | | | | | | | | | |
| CoT SFT + GRPO | | ✓ | ✓ | 104\|104\|50 | 47.05 | 64.50 | 76.42 | 68.02 | 72.22 |
| CoT SFT + DAPO | | ✓ | ✓ | 104\|85*\|50 | 67.60 | 74.40 | 76.42 | 68.44 | 72.43 |
| *TriMPI without PolicyRollout (PoRo)* | | | | | | | | | |
| TriMPI w/ GRPO | ✓ | ✓ | ✓ | 104\|104\|50 | 55.90 | 80.80 | 83.96 | 74.70 | 79.33 |
| TriMPI w/ DAPO | ✓ | ✓ | ✓ | 70*\|70*\|50 | 65.85 | 81.45 | 79.25 | 70.85 | 75.05 |
| **TriMPI with PolicyRollout (PoRo)** | | | | | | | | | |
| **TriMPI w/ PoRo-GRPO** | ✓ | ✓ | ✓ | 104\|104\|50 | 65.85 | 84.70 | **85.85** | **76.28** | **81.06** |
| **TriMPI w/ PoRo-DAPO** | ✓ | ✓ | ✓ | 90*\|75*\|50 | **77.80** | **85.00** | 80.19 | 71.82 | 76.01 |

$$\mathcal{J}_{\text{PoRo-GRPO}}(\theta) = \mathbb{E}_{[\{o_i\}_{i=1}^{G} \sim \pi_{\theta_{old}}(O|Q,I), \ \{o_j\}_{j=G}^{2G} \sim \pi_{\theta_{old}}(O|Q,I,P)]} \frac{1}{2G} \sum_{i=1}^{2G} \Big\{$$

$$\min \Big[ r_i(\theta)\hat{A}_i, \text{clip}\left(r_i(\theta), 1 - \epsilon_l, 1 + \epsilon_h\right) \hat{A}_i \Big] - \beta \mathbb{D}_{KL} \left[\pi_\theta || \pi_{ref}\right] \Big\}, \ r_i(\theta) = \frac{\pi_\theta(o_i|Q,I)}{\pi_{\theta_{old}}(o_i|Q,I)} \quad (3)$$

The blue part highlights the main modification compared with the original GRPO objective. Note that the policy gradient is applied only to the no-policy path (conditioning solely on $Q$ and $I$), thereby ensuring that the training and inference remain aligned.

## 5 EXPERIMENTS

We conduct comprehensive experiments on multimodal policy internalization, evaluating **MPI task performance** (§5.1), **generalization capability** (§5.2), **policy knowledge injection** (§5.3), and **robustness to catastrophic forgetting** (§5.5). By default, we use Qwen2.5-VL-7B as the base model. We tune all model parameters in the VM-CPT and RL stages, and apply LoRA (Hu et al., 2022) finetuning in the SFT stage. For ClevrPolicy, we use the most complex policies ($N = 6$) unless otherwise specified. Additional implementation details are provided in Appendix B.

### 5.1 MAIN RESULTS

**MPI Task Performance.** In **Table 2**, we present evaluation results on ClevrPolicy and GTAPolicy in terms of task performance. Our best-performing model achieves up to 70.7% and 79.4% absolute gains in accuracy over the CoT SFT baseline and the in-context setting, respectively. We also find that the relative superiority of GRPO and DAPO varies across datasets: DAPO performs better on ClevrPolicy, while GRPO excels on GTAPolicy. We hypothesize that this difference arises because DAPO makes bolder updates due to the removal of the reference KL. This leads to faster learning on ClevrPolicy, which contains more abundant and diverse data, but results in overfitting on GTAPolicy, which has very limited data. We further present a qualitative example in **Figure 7** in the Appendix,

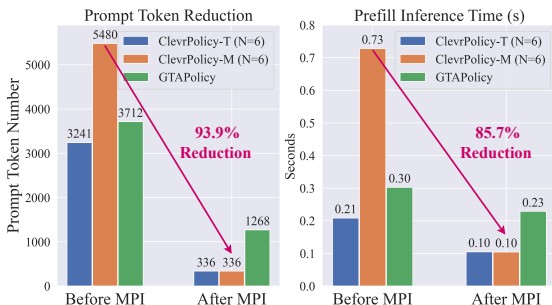

Figure 6: **Efficiency** metrics before and after MPI.

Table 3: **Left: Policy Override results.** We show that TriMPI consistently outperforms strong baselines in generalizing to updated policies, demonstrating favorable real-world usage where model behavior can be governed by both internalized policies and in-context instructions. **Right: Policy Referral results.** We use Claude-4 to rank the consistency between the model's intermediate thoughts and the original policy on a scale of 0-10. A higher score indicates better embedded policy knowledge.

| Method | ClevrPolicy Override (N=6) | | GTAPolicy Override | | | Policy Referral (0-10 Scores) | | |
|---|---|---|---|---|---|---|---|---|
| | ClevrPolicy-T | ClevrPolicy-M | Tool Acc | Arg Score | Overall | ClevrPolicy-T | ClevrPolicy-M | GTAPolicy |
| In-Context | 13.15 | 6.55 | 20.75 | 18.11 | 19.43 | - | - | - |
| Direct SFT | 5.40 | 5.30 | 40.57 | 33.42 | 36.99 | - | - | - |
| CoT SFT | 16.40 | 25.20 | 42.45 | 34.98 | 38.71 | 3.54 | 3.44 | 7.68 |
| CoT SFT + GRPO | 30.80 | 34.00 | 50.94 | 41.60 | 46.27 | 5.04 | 6.74 | 8.93 |
| CoT SFT + DAPO | 41.60 | 37.60 | 58.49 | 50.12 | 53.31 | 5.70 | 6.74 | 8.73 |
| TriMPI w/ PoRo-GRPO | 48.70 | 82.70 | **66.98** | **59.03** | **63.00** | 5.60 | **8.72** | **9.45** |
| TriMPI w/ PoRo-DAPO | **59.40** | **85.15** | 63.21 | 54.66 | 58.94 | **6.26** | 8.35 | 9.13 |

showing that TriMPI achieves better alignment with the policy in both intermediate thoughts and final answers.

**Ablation Analysis.** In **Table 2**, we additionally provide a comprehensive ablation study on the key components proposed in TriMPI, including the RL stage, the VM-CPT stage, and the PolicyRollout algorithm. The findings are as follows: (1) The RL stage contributes most of the improvements compared with SFT baselines, highlighting the importance of learning from experience for complex, reasoning-intensive policies. We also observe a unique benefit of the RL stage in better leveraging non-CoT data, which is typically more abundant than CoT data, as shown in **Table 11** in the Appendix. (2) the VM-CPT stage brings further improvements to both the SFT and RL stages, with the effect being more pronounced in the RL stage due to more grounded exploration; and (3) PolicyRollout yields additional gains over the original GRPO and DAPO algorithms.

**Efficiency Analysis.** We report efficiency metrics in Figure 6, including the number of prompt tokens and the prefill inference time (i.e., the first forward pass on the input prompt) before and after internalization. All metrics are computed on Qwen2.5-VL-7B. With the policy removed from the prompt, we observe reductions of up to 93.9% in prompt tokens and 85.7% in prefill inference time.

**Impact of Policy Complexity and Model Size.** We further investigate the impact of policy complexity and model size on the effectiveness of different MPI algorithms. As shown in **Table 8** in the Appendix, we conduct additional experiments on (1) $N = 4$ policies in ClevrPolicy and (2) 3B models. Our findings are twofold: (1) TriMPI yields more pronounced gains over the baselines on complex policies, while the performance gap is smaller on simpler policies (e.g., $N = 4$). This highlights the importance of our method for handling increasingly complex policies in real-world settings. (2) TriMPI consistently outperforms the baselines on 3B models, demonstrating the generality of the method.

## 5.2 POLICY OVERRIDE: EVALUATING GENERALIZATION TO POLICY UPDATES

In real-world scenarios, policies may be frequently updated or partially overridden by new rules. Ideally, the internalized model should generalize to these updated policies when they are provided in-context. To evaluate this capability, we introduce a new evaluation setting, *Policy Override*,

Table 4: **Policy In-Context results.** We evaluate models after MPI training with the policies placed back into the context. This aims to demonstrate the effectiveness of the proposed methods regardless of the requirement to remove policy inputs during inference for efficiency. We find that TriMPI still achieves superior performance under this setting. The base model is Qwen2.5-VL-7B.

| Method | ClevrPolicy (N=6) | | GTAPolicy | | |
|---|---|---|---|---|---|
| | ClevrPolicy-T | ClevrPolicy-M | Tool Acc | Arg Score | Overall |
| In-Context (no MPI) | 13.15 | 5.65 | 23.58 | 19.44 | 21.51 |
| CoT SFT | 10.70 | 15.80 | 45.28 | 39.24 | 42.26 |
| CoT SFT + GRPO | 19.00 | 20.90 | 60.38 | 53.64 | 57.01 |
| TriMPI w/ PoRo-GRPO | **35.15** | **69.90** | **70.75** | **62.86** | **66.81** |

where unseen policy content is specified in-context during inference with internalized models. As illustrated in **Figure 10** in the Appendix, for ClevrPolicy we retain the policy name but modify the conditions, while for GTAPolicy we alter the tool-calling rules, resulting in a different choice of tool version. **Table 3 (left)** shows that TriMPI consistently outperforms all baselines and ablated settings, demonstrating stronger generalization beyond merely fitting to a specific policy.

### 5.3 POLICY REFERRAL: EVALUATING POLICY KNOWLEDGE INJECTION

Beyond end-task performance, which measures how well the model generates policy-compliant responses, we further investigate how well the model embeds the policy knowledge itself. To this end, we consider a new evaluation setting, *Policy Referral*, where we leverage LLM-as-a-judge to examine the intermediate reasoning process of the model's responses. The goal is to assess whether the referral to the original policy is accurate and to assign a score between 0 and 1. **Figure 8** in the Appendix illustrates this evaluation setting, and **Table 3 (right)** presents the results, reporting the average score over a subset of 100 test responses from each dataset. We find that TriMPI achieves more accurate policy referral, indicating that it not only learns end-task behavior but also internalizes the underlying policy.

### 5.4 POLICY IN-CONTEXT: EVALUATING DISTINCTIVE IMPACT ON POLICY FOLLOWING

Consider the additional training stage and the rollout computation overhead in TriMPI, an potential alternative is to use baseline internalization methods but put the policy back in-context during inference. In this section, we aim to examine if the TriMPI still offers better performance when given the policy in-context during inference. As shown in **Table 4**, TriMPI consistently outperforms baselines under this setting.

### 5.5 EVALUATING ROBUSTNESS TO CATASTROPHIC FORGETTING

Similar to related work in continual learning (Yu et al., 2024) and knowledge injection (Song et al., 2025), a major challenge in MPI is avoiding catastrophic forgetting, ensuring that policy-related performance improves while general non-policy abilities are preserved. To assess this, we adopt two widely used general reasoning benchmarks, MMMU-Pro (Yue et al., 2024) (multimodal) and MMLU-Pro (Wang et al., 2024b) (textual), to evaluate the model's robustness to catastrophic forgetting after internalization. The results are presented in **Table 9** in the Appendix. We observe that the baselines exhibit significant performance degradation after MPI on GTAPolicy, while maintaining strong performance after MPI on ClevrPolicy. This discrepancy arises because the small dataset size of GTAPolicy makes it more prone to overfitting. In contrast, TriMPI consistently preserves strong general reasoning abilities across all settings, achieving the best overall performance. In Appendix H, we further evaluate on WildGuardTest (Han et al., 2024) to examine the model's safety-related performance.

### 5.6 ERROR ANALYSIS

**Qualitative Failure Mode Breakdown.** We further provide a breakdown of failure modes on our best-performing model to highlight remaining challenges. On ClevrPolicy, we observe both perception and reasoning errors. Perception errors include missing occluded objects and misidentifying

attributes in scenes with multiple similar objects. Reasoning errors primarily involve branching to non-existent Conditions or hallucinating the rules even when the correct section is chosen. GTAPolicy errors are largely reasoning-driven due to its simpler visual scenes. A typical issue is misusing interaction history, e.g., incorrectly calling an image-level description tool instead of a region-specific one after obtaining a bounding box. Qualitative error examples are visualized in Appendix M.

**Quantitative Analysis on the Branching Error.** Branching error is one of the major reasoning errors on ClevrPolicy, where the model hallucinates a Condition section that does not exist in the target policy, indicating overfitting and insufficient grounding to the policy. To quantitatively analyze whether the proposed TriMPI effectively reduces this error, we compare the distribution of referenced Condition IDs in model-generated responses against those in a synthesized Gold CoT response. As shown in **Figure 14** in Appendix, we find that GRPO does not meaningfully reduce the discrepancy in Condition ID distribution relative to the SFT baseline (Average Absolute Difference: 38.72 vs. 39.74), indicating that GRPO alone does not sufficiently improve referral accuracy to the policy. In contrast, TriMPI achieves a substantial reduction in this discrepancy (Average Absolute Difference: 7.07), demonstrating that TriMPI enables more grounded and policy-aligned reasoning.

## 6 RELATED WORK

### 6.1 PROMPT COMPRESSION AND DELIBERATIVE ALIGNMENT

A line of related research (Li et al., 2024) focuses on compressing long prompts into more compact forms that can still effectively guide large language models. Early efforts explored both hard prompts (Li et al., 2023; Jiang et al., 2023; Chuang et al., 2024), where discrete tokens are carefully pruned, and soft prompts (Zhao et al., 2023; Wingate et al., 2022; Mu et al., 2023; Ge et al., 2023), where continuous embeddings are learned to replace verbose instructions or demonstrations. More recent work, such as PromptIntern (Zou et al., 2024), adopts a progressive fine-tuning approach that internalizes prompts into the model without introducing additional parameters. Another line of related work is on personalized multimodal models (Nguyen et al., 2024; 2025), for which we provide a detailed discussion in Appendix L. While promising, these methods focus on prompts limited to task templates and demonstrations that demand little reasoning. Moreover, the introduction of special embeddings confines the model to a specific task, reducing its ability to handle general queries.

Deliberative Alignment (Guan et al., 2024; Zhang et al., 2025) extends this idea to a more general alignment setting without sacrificing the model's overall capabilities. Its goal is to embed the knowledge of a safety specification into the model parameters while emphasizing reasoning over the internalized knowledge. Our proposed MPI task follows this high-level motivation but further extends the scope to multimodal models and considers diverse conversational agent tasks beyond safety.

### 6.2 IMPROVING GRPO FROM THE ROLLOUT PERSPECTIVE

We also draw inspiration from recent multimodal reasoning work that investigates improving GRPO-related RLVR algorithms from the rollout perspective. Methods such as NoisyRollout (Liu et al., 2025) and R1-ShareVL (Yao et al., 2025) demonstrate the benefits of diversifying the rollout space with responses generated from altered instances using semantically consistent augmentations, such as applying moderate Gaussian noise to the visual inputs. Our proposed PolicyRollout algorithm follows the same high-level idea of augmenting the rollout space, but instead of introducing noise, it provides the policy in-context to enable more grounded exploration. PolicyRollout illustrates the potential of a more general approach to incorporating additional guidance in the RL stage, while remaining aligned with the original optimization task.

## 7 CONCLUSION AND LIMITATIONS

In this work, we propose a new task, Multimodal Policy Internalization, which aims to address the emerging challenge of maintaining in-context efficiency while following complex policies in multimodal conversational agents. We introduce two new benchmarks spanning analytical and real-world settings, along with a highly effective training paradigm, TriMPI. The remaining limitations are: (1) scaling up the datasets with more diverse real-world images and tasks; (2) developing more sophisticated continual pretraining strategies beyond simply masking visual tokens; and (3) designing training strategies for internalizing mixtures of tasks with very different response formats.

## 8 REPRODUCIBILITY STATEMENT

We include an anonymous source code archive in the supplementary material, containing training and evaluation instructions for reproducing the results in this paper. Details of dataset creation and evaluation metrics are provided in §C and §D. Full policy examples are shown in Figures 18, 19, and 20. Implementation details of the training procedure are provided in §B. Prompts used for the LLM-as-a-judge Policy Referral evaluation are presented in Figure 9.

## 9 ETHICS STATEMENT

This work focuses on fundamental research aimed at advancing the understanding of how multimodal models can internalize complex policies more effectively. All experiments are conducted on publicly available datasets, and no human subjects or private user data are involved. The GTAPolicy dataset introduced in this work contains fully synthesized user profiles and does not include or rely on any real user data. Code and data from this work will be made publicly available for research purposes in the near future.

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

APPENDIX

Table 5: **MPI training hyperparameters** for different stages. EP denotes the number of epochs, LR is the learning rate, and BS is the batch size, expressed as per-device batch size × number of devices. For the RL stage, BS refers to the update batch size. Additional configuration details, such as the rollout batch size, are provided in Appendix B. Table 6 shows the actually RL steps taken in different training settings. Table 7 shows the exact dataset sizes.

| Stage | Dataset | Data Type | Data Size | LoRA | LoRA Rank | EP | LR | BS |
|---|---|---|---|---|---|---|---|---|
| VM-CPT | ClevrPolicy | Policy + CoT | 2.5K | ✗ | - | 5 | 1e-5 | 16×8 |
| VM-CPT | GTAPolicy | Policy + CoT | 0.5K | ✗ | - | 5 | 1e-5 | 8×8 |
| Direct SFT | ClevrPolicy | Non-CoT | 20K | ✓ | 8 | 2 | 1e-4 | 16×4 |
| Direct SFT | GTAPolicy | Non-CoT | 0.5K | ✓ | 8 | 10 | 1e-4 | 8×4 |
| CoT SFT | ClevrPolicy | CoT | 2.5K | ✓ | 8 | 5 | 1e-4 | 16×4 |
| CoT SFT | GTAPolicy | CoT | 0.5K | ✓ | 8 | 10 | 1e-4 | 8×4 |
| RL | ClevrPolicy | Non-CoT | 20K | ✗ | - | 2 | 1e-6 | 4×8 |
| RL | GTAPolicy | Non-CoT | 0.5K | ✗ | - | 50 | 1e-6 | 4×8 |

## A    USE OF LARGE LANGUAGE MODELS

Large language models, such as ChatGPT and Claude, are used solely for grammar checking during the writing process and not for research ideation. Additionally, LLM-based coding assistants, such as Copilot, are employed to aid in code implementation.

## B    IMPLEMENTATION DETAILS

**In-context Experiment Details.**    The exact model versions used in Table 1 are as follows. For Qwen2.5-VL, we use the model checkpoints from Huggingface (3B[†], 7B[‡]). For Claude models, we use "claude-3-7-sonnet-20250219" and "claude-sonnet-4-20250514" hosted on Amazon Bedrock[§].

**Multimodal Policy Internalization Training Details.**    Table 5 summarizes the training hyperparameters for different stages. For the RL stages, we set the clipping ratio to $\epsilon_l = 0.2, \epsilon_h = 0.3$ for GRPO and $\epsilon_l = 0.2, \epsilon_h = 0.28$ for DAPO. The rollout batch size is 384 for ClevrPolicy and 256 for GTAPolicy. The reference KL coefficient is set to $\beta = 0.01$ for GRPO. The maximum number of retries for dynamic sampling is set to 20 for DAPO. The actual number of RL steps taken in each run is reported in Table 6. We use 4 NVIDIA H100 80GB GPUs for SFT stages and 8 H100 for CPT and RL stages.

## C    DATASET CREATION DETAILS

### C.1    CLEVRPOLICY

#### C.1.1    POLICY GENERATION IN CLEVRPOLICY

As described in §3.1, we first generate a set of binary decision trees to construct the policies. We define two types of nodes: decision nodes and response nodes. A decision node checks whether the current input image satisfies a sampled condition (for example, the existence of a cyan object), while a response node corresponds to a unique outcome. Each decision node (non-leaf node) samples an attribute type and an attribute value from the following ontology:

- **Shape**: "cube", "sphere", "cylinder"
- **Color**: "gray", "red", "blue", "green", "brown", "purple", "cyan", "yellow"
- **Size**: "small", "large"

---

[†] https://huggingface.co/Qwen/Qwen2.5-VL-3B-Instruct
[‡] https://huggingface.co/Qwen/Qwen2.5-VL-7B-Instruct
[§] https://docs.anthropic.com/en/api/claude-on-amazon-bedrock

Table 6: **Detailed statistics of the number of RL steps** under different settings. Numbers annotated with (*) denote early stopping, which may occur in DAPO runs due to dynamic sampling.

| Methods | RL Steps | | | | |
|---|---|---|---|---|---|
| | ClevrPolicy-T (N=6) | ClevrPolicy-M (N=6) | ClevrPolicy-T (N=4) | ClevrPolicy-M (N=4) | GTAPolicy |
| Base Model: Qwen2.5-VL-7B | | | | | |
| CoT SFT + GRPO | 104 | 104 | 104 | 104 | 50 |
| CoT SFT + DAPO | 104 | 85* | 25* | 45* | 50 |
| TriMPI w/ PoRo-GRPO | 104 | 104 | 104 | 104 | 50 |
| TriMPI w/ PoRo-DAPO | 90* | 75* | 15* | 20* | 50 |
| Base Model: Qwen2.5-VL-3B | | | | | |
| CoT SFT + GRPO | 104 | 104 | - | - | - |
| CoT SFT + DAPO | 104 | 104 | - | - | - |
| TriMPI w/ PoRo-GRPO | 104 | 104 | - | - | - |
| TriMPI w/ PoRo-DAPO | 100* | 95* | - | - | - |

- **Material**:"rubber", "metal"

If a child decision node has a parent node and the edge to the parent node is true, the child node is restricted from sampling the same attribute type. For example, if the parent node checks for cyan objects and the edge is true, the child node will not check the color attribute again, to avoid logical conflicts. This results in 4, 16, and 55 unique target responses for policies with $N = 2$, $N = 4$, and $N = 6$, respectively.

Given a sampled decision tree, we then convert it into a textual instruction containing three main components: a general instruction, a unique policy name, and sections of conditions governing the response behavior. This policy challenges the model to perform multi-hop reasoning across different sections in order to faithfully follow the rules.

We further introduce two variants of ClevrPolicy, namely, ClevrPolicy-T and ClevrPolicy-M, depending on whether the policy itself contains image content. As shown in Figure 2 (left), in ClevrPolicy-M the attribute value in a decision node may be demonstrated by a small image crop instead of natural language. This introduces an additional level of difficulty for policy following.

### C.1.2 TRAINING DATA CREATION IN CLEVRPOLICY

We generate a mixture of 10 unique policies for each of the three levels of complexity indicated by the layer number $N \in 2, 4, 6$. For each policy, we then sample 2K images as training data. The ground-truth answers can be automatically obtained by verifying the scene graph against the corresponding decision tree. This results in 20K non-CoT QA pairs for each training task in ClevrPolicy-T and ClevrPolicy-M. Similarly, we sample another 2K unseen instances for testing. Importantly, the underlying decision trees are **not** accessible to the model during subsequent MPI training. An input-output example is illustrated in Figure 2 (right). Evaluation is performed based on exact string matching, and results are reported in terms of accuracy.

### C.2 GTAPOLICY

### C.2.1 POLICY CREATION IN GTAPOLICY

We manually construct a complex policy based on the GTA dataset (Wang et al., 2024a), which was originally proposed as a multimodal tool using benchmark. As illustrated in Figure 3 (left), the GTAPolicy policy consists of two main components: tool descriptions and tool calling rules. The tool descriptions contain the metadata for all 13 types of tools, including the tool name, description, and arguments. To ensure that model performance reflects its ability to follow the internalized policy rather than relying on prior knowledge, we further introduce a *versioning mechanism* and *user-conditional tool-calling rules*. Specifically, a tool such as "OCR" may have multiple versions, e.g., "OCR_v1" and "OCR_v2." A total of 24 tool-calling rules specify which particular version to use depending on the profile attributes of the input user. This design simulates real-world business

rules in which certain user properties must be considered during decision-making and tool calling, such as whether the user is a premium member.

### C.2.2 TRAINING DATA CREATION IN GTAPOLICY

We first reformulate the multi-turn data from the GTA dataset into a single-turn tool calling task to avoid overcomplicating the MPI setting with intermediate tool calling errors. As illustrated in Figure 3 (right), each input instance consists of five parts: visual inputs (potentially multiple images), a user profile, a user query, an interaction history containing previous tool calls and returned values, and a general instruction prompt. The expected output is a tool call formatted in JSON that specifies the tool name (with version) and the corresponding arguments. The final dataset contains 451 instances for training and 106 instances for testing. To evaluate the tool calls, we use exact match on the tool name, and apply different evaluation metrics to the argument values depending on their type (e.g., IoU for bounding boxes, text similarity for free form text queries). Detailed evaluation metrics are provided in Appendix C.3.

### C.3 EVALUATION METRICS AND RLVR REWARDS

**Evaluation Metrics for ClevrPolicy.** We use accuracy based on the exact match between the predicted and ground truth outcomes, for example, "Case 0". Accordingly, the following accuracy reward is used for GRPO and DAPO on ClevrPolicy:

$$R_{\text{Acc-ClevrPolicy}} = \mathbf{1}[\text{Exact\_Match}(y, \hat{y})], \tag{4}$$

where $y$ and $\hat{y}$ denote the parsed response and the ground truth, respectively.

**Evaluation Metrics for GTAPolicy.** For the tool name, we use accuracy based on exact match. For the arguments, we identify four categories of argument types and apply different metrics accordingly. The full list of tools and argument definitions is shown in Figure 20. We provide the argument names and their corresponding evaluation metrics as follows:

- **Exact match**: "position", "color", "image", "k", "top1"

- **Text similarity**: "attribute", "text", "query", "command", "annotation", "keywords", "instruction"

- **Python Eval**: "expression"

- **IoU**: "bbox"

Exact match is used for arguments that require precise string matching, for example, the image name when running a tool. The text similarity metric evaluates arguments in free-form text format, such as search queries or object names. We use BertScore (Zhang et al., 2019) to compute the semantic similarity between two text sequences. For expressions involving numerical operations, we use the Python "eval" function to compute the final outcome and compare it with the ground truth. For bounding box arguments, we use Intersection over Union (IoU) to compute the overlap between the predicted and ground truth coordinates.

All scores are normalized to the range $[0, 1]$. The final argument score is computed as the average across all argument fields. We then compute an overall score for the entire tool call as the average of the tool accuracy and the argument score. Accordingly, the following accuracy reward is used for GRPO and DAPO on GTAPolicy:

$$R_{\text{Acc-GTAPolicy}} = 0.5 \times \text{Tool\_Acc} + 0.5 \times \text{Argument\_Score} \tag{5}$$

### C.4 FULL POLICY EXAMPLES

Examples of the full policy for ClevrPolicy-T, ClevrPolicy-M and GTAPolicy are presented in Figures 18, 19 and 20, respectively.

Table 7: **Detailed dataset statistics.**

| Dataset | CoT Generator | CoT Strategy | Filtering | Train Size CoT | Non-CoT | Test Size |
|---|---|---|---|---|---|---|
| ClevrPolicy-T (N=6) | Qwen2.5-VL-7B | Forward CoT | Yes | 2526 | 20000 | 2000 |
| ClevrPolicy-T (N=4) | Qwen2.5-VL-7B | Forward CoT | Yes | 2526 | 20000 | 2000 |
| ClevrPolicy-M (N=6) | Claude-4-Sonnet | Forward CoT | Yes | 2472 | 20000 | 2000 |
| ClevrPolicy-M (N=4) | Claude-4-Sonnet | Forward CoT | Yes | 2282 | 20000 | 2000 |
| GTAPolicy | Claude-4-Sonnet | Reverse CoT | No | 451 | 451 | 106 |

Table 8: **Additional results on varying task complexity and model size.** We show that TriMPI consistently outperforms strong baselines across different complexities and model sizes. Notably, the performance gain is more pronounced on complex policies.

| Method | Varying Policy Complexity ClevrPolicy-T N=4 | N=6 | ClevrPolicy-M N=4 | N=6 | Varying Model Size ClevrPolicy-T (N=6) 3B | 7B | ClevrPolicy-M (N=6) 3B | 7B |
|---|---|---|---|---|---|---|---|---|
| In-Context | 32.20 | 13.15 | 13.90 | 5.65 | 4.80 | 13.15 | 4.55 | 5.65 |
| Direct SFT | 25.65 | 15.15 | 37.85 | 14.55 | 13.40 | 15.15 | 12.90 | 14.55 |
| CoT SFT | 49.10 | 17.80 | 55.35 | 14.30 | 19.00 | 17.80 | 9.25 | 14.30 |
| CoT SFT + GRPO | 98.25 | 47.05 | 87.10 | 64.50 | 31.55 | 47.05 | 22.40 | 64.50 |
| CoT SFT + DAPO | 97.15 | 67.60 | 88.75 | 74.40 | 48.10 | 67.60 | 33.25 | 74.40 |
| TriMPI w/ PoRo-GRPO | **99.15** | 65.85 | 99.05 | 84.70 | 46.65 | 65.85 | 78.80 | 84.70 |
| TriMPI w/ PoRo-DAPO | 97.30 | **77.80** | **99.40** | **85.00** | **68.75** | **77.80** | **82.35** | **85.00** |

# D DETAILS ON COT SFT

## D.1 COT DATA GENERATION

Given the practical challenges of accessing APIs for stronger models such as GPT and Claude, a key principle we follow is to minimize reliance on API models whenever possible.

Specifically, on ClevrPolicy, we use **forward CoT** with answer filtering to generate CoT data, where a generator model is asked to solve the task with intermediate reasoning steps given the policy and the input query. The generated instances are then filtered based on answer correctness. For GTAPolicy, due to the small dataset size, we adopt **reverse CoT**, where the generator model is provided with both the inputs and the ground truth answer and asked to generate the intermediate rationales. Reverse CoT is suitable in such settings where filtering is difficult or the data samples are scarce.

We select the generator model based on the zero-shot in-context performance shown in Table 1. For ClevrPolicy-T, we use the base model itself, Qwen-2.5VL-7B (Bai et al., 2025), as the generator. For more challenging settings, i.e., ClevrPolicy-M and GTAPolicy, we use Claude-4(Anthropic, 2025). In the case of ClevrPolicy-M with Claude-4, we subsample the input non-CoT data to 3K in order to match the CoT dataset size of ClevrPolicy-T. After filtering, we obtain approximately 2.5K CoT instances for each ClevrPolicy setting and 451 instances for GTAPolicy. The full statistics of the CoT and non-CoT data can be found in **Table 7**. **Figures 15, 16 and 17** provide examples of the CoT data for all three datasets.

## D.2 COT SFT OBJECTIVE

Given the generated CoT annotation $C$ for each question-answer pair $(Q, A)$, the training objective for the CoT SFT stage can be written as follows:

$$\mathcal{L}_{\text{SFT}} = -\mathbb{E}_{(Q,O)\sim\mathcal{D}} \left[ \sum_{t=1}^{|O|} \log p_\theta(o_t \mid Q, o_{<t}) \right], O = [C; A] \quad (6)$$

where $O$ is a concatenation of $C$ and $A$.

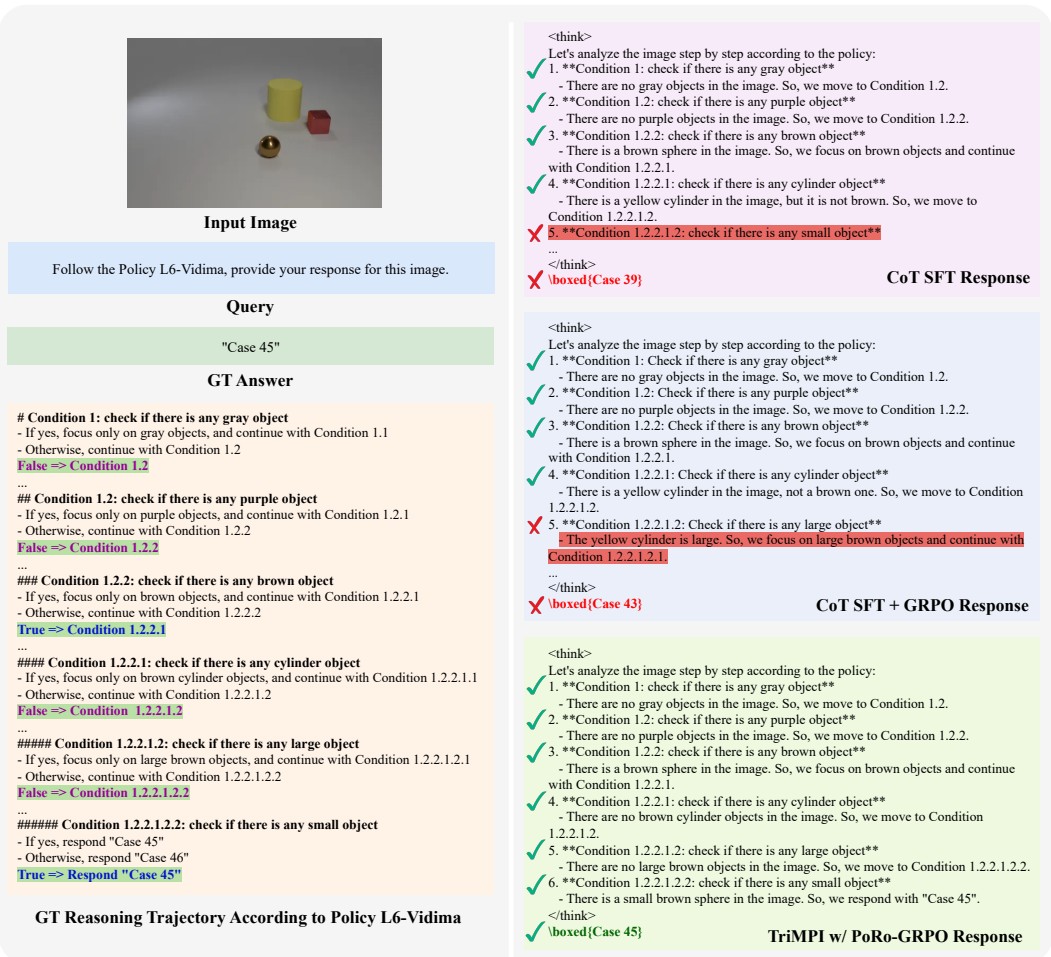

Figure 7: **Qualitative example comparing different MPI algorithms.** On the left, we show the inputs and the ground-truth reasoning trajectory annotated with the original policy sections. On the right, the CoT SFT model makes an error in correctly recalling the policy condition, and the CoT SFT + GRPO model makes an incorrect decision at the fifth condition, both leading to an incorrect final outcome. In contrast, the proposed TriMPI correctly recalls all policy conditions and performs reasoning consistent with the input image.

# E    ADDITIONAL RESULTS ON VARYING POLICY COMPLEXITY AND MODEL SIZE

Table 8 illustrates the additional results on varying task complexity and model size. TriMPI is especially valuable for handling complex policies, showing larger gains over baselines in these settings while maintaining consistent improvements even on simpler cases (e.g., $N = 4$). Moreover, it generalizes well across model scales, consistently outperforming baselines on 3B models as well.

# F    QUALITATIVE ANALYSIS

Figure 7 shows a qualitative example from the ClevrPolicy dataset, comparing the responses of different MPI algorithms. The results demonstrate that the proposed TriMPI algorithm successfully refers to the original policy and produces a policy-compliant answer, whereas the SFT and SFT+RL baselines suffer from incorrect policy referral or reasoning errors.

Table 9: **Robustness to catastrophic forgetting on general reasoning.** We evaluate the internalized models on MMMU-pro (Yue et al., 2024) and MMLU-pro (Wang et al., 2024b) benchmarks. The results show that TriMPI consistently maintains strong general capabilities across all settings, while most baselines exhibit a significant drop on the smaller dataset, i.e., GTAPolicy.

| Method | MMMU-Pro (Acc) | | | MMLU-Pro (Acc) | | | AVG |
|---|---|---|---|---|---|---|---|
| | ClevrPolicy-T | ClevrPolicy-M | GTAPolicy | ClevrPolicy-T | ClevrPolicy-M | GTAPolicy | |
| Before MPI Training | | | | | | | |
| Qwen2.5-VL-7B | 31.91 | 31.91 | 31.91 | 31.73 | 31.73 | 31.73 | 31.82 |
| After MPI Training | | | | | | | |
| Direct SFT | 34.10 | 33.24 | 27.86 | **39.25** | 37.01 | 28.91 | 33.40 |
| CoT SFT | 31.85 | 32.95 | 10.17 | 37.67 | 38.70 | 10.43 | 26.96 |
| CoT SFT + GRPO | 32.60 | 32.08 | 22.95 | 37.63 | 40.13 | 22.66 | 31.34 |
| CoT SFT + DAPO | **34.22** | **34.16** | 18.84 | 38.41 | **39.79** | 20.86 | 31.04 |
| TriMPI w/ PoRo-GRPO | 31.45 | 33.29 | 30.12 | 38.57 | 39.34 | **35.77** | **34.76** |
| TriMPI w/ PoRo-DAPO | 31.56 | 32.66 | **30.52** | 37.68 | 39.49 | 35.63 | 34.59 |

Table 10: **Robustness to catastrophic forgetting on safety.** We evaluate the internalized models on WildGuardTest (Han et al., 2024). The results show that TriMPI achieves more consistent preservation of safety-related performance compared with other baselines.

| Method | WildGuardTest (Acc) | | | AVG |
|---|---|---|---|---|
| | ClevrPolicy-T | ClevrPolicy-M | GTAPolicy | |
| Before MPI Training | | | | |
| Qwen2.5-VL-7B | **87.24** | **87.24** | **87.24** | **87.24** |
| After MPI Training | | | | |
| CoT SFT | 82.04 | 52.95 | 69.22 | 68.07 |
| CoT SFT + GRPO | 83.21 | 47.69 | 69.87 | 66.92 |
| TriMPI w/ PoRo-GRPO | 74.90 | 76.71 | 76.77 | 76.13 |

## G ROBUSTNESS TO CATASTROPHIC FORGETTING ON GENERAL REASONING

Table 9 presents results on general multimodal (MMMU-pro (Yue et al., 2024)) and textual (MMLU-pro (Wang et al., 2024b)) reasoning benchmarks for models after MPI training. This evaluation examines whether the model preserves general reasoning capabilities while internalizing policy-compliant behavior.

## H ROBUSTNESS TO CATASTROPHIC FORGETTING ON SAFETY

To further investigate the preservation of safety-related behavior, we include an additional benchmark, WildGuardTest (Han et al., 2024). Given a prompt and a model response, the task is to determine whether the response is harmful (e.g., involving privacy violations, misinformation, harmful language, or malicious uses). Table 10 reports the accuracy (%) on WildGuardTest for models after training with ClevrPolicy-T, ClevrPolicy-M, and GTAPolicy. We show that the proposed TriMPI method maintains the most consistent performance across different policy-internalization settings, preserving most of the overall capabilities. In contrast, the SFT and GRPO baselines exhibit larger variance and, in particular, suffer a substantial performance drop under the ClevrPolicy-M setting.

## I ILLUSTRATION OF THE POLICY REFERRAL EVALUATION

Figure 8 illustrates the Policy Referral evaluation setting, where we leverage a strong LLM, Claude-4 (Anthropic, 2025), to score how accurate does the response cited to the original policy. The exact prompt we used for prompting the LLM is presented in Figure 9. This evaluation aims to test whether the model has actually absorbed the policy knowledge beyond merely mimicking the expected behavior.

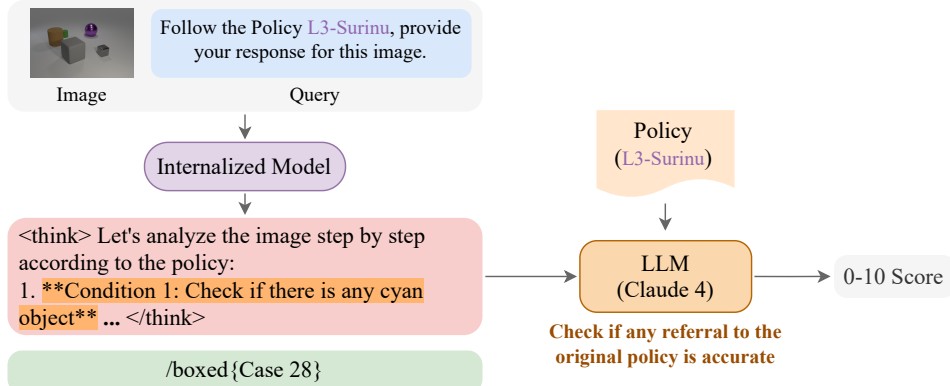

Figure 8: **Illustration of the Policy Referral evaluation setup.** We take the responses from the internalized model and ask a strong LLM (Claude-4 (Anthropic, 2025)) to score the consistency between any policy referral in the response and the original policy. Policy referral is designed to evaluate the quality of the embedded policy knowledge beyond end-task performance.

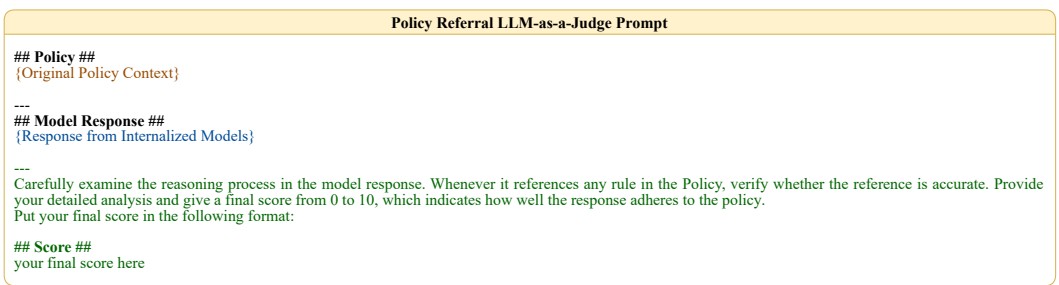

Figure 9: **Prompt to Claude-4 for the Policy Referral evaluation.**

## J    ILLUSTRATION OF THE POLICY OVERRIDE EVALUATION

Figure 10 illustrates the Policy Override evaluation setting, where we place the updated policy in-context during evaluation and expect the model to follow the overridden policy. This evaluation aims to test the model's ability to generalize in policy following, beyond memorizing a particular policy.

## K    RL EFFECTIVELY LEVERAGES NON-COT DATA

During CoT data generation, we observe that in real-world settings, non-CoT data is often much more abundant than CoT data. For example, in ClevrPolicy, after filtering, we obtain 2.5K CoT examples versus 20K non-CoT examples. A unique benefit of the RL stage (with GRPO and DAPO) is that it does not require CoT annotations. To investigate the extent to which the RL stage can benefit from non-CoT data relative to CoT data, we conduct the following experiment. We run RL algorithms using only CoT data and additionally include an SFT baseline, where the model is first trained on CoT data and then further trained on non-CoT data. The results are shown in Table 11. We observe that the RL stage can more effectively leverage the abundant non-CoT data compared to SFT-only approaches.

## L    ADDITIONAL RELATED WORK

Another related line of work from the multimodal community is personalization through textual inversion (Gal et al., 2022). This spans from early studies on personalized image generation (Gal et al., 2022; Ruiz et al., 2023) to more recent approaches on personalized multimodal models (Nguyen et al., 2024; Alaluf et al., 2024; Nguyen et al., 2025). The central idea is to learn lightweight embeddings

Figure 10: **Illustration of the Policy Override evaluation setup.** For ClevrPolicy, we provide a randomly sampled new policy in-context during inference as updated policy content while keeping its original unique name. For GTAPolicy, we modify the tool-calling rules, resulting in a different model version. Policy Override evaluates the model's capability to generalize to updated or modified policies beyond overfitting. This property is particularly important in real-world scenarios, where policies are constantly changing and model behavior must be governed by both the internalized policy and the in-context instructions.

Table 11: **Results of the RL stage with different data source.** RL algorithms such as GRPO can better leverage the more abundant non-CoT data compared to SFT.

| Method | Data | | Dataset Size | | ClevrPolicy-T | ClevrPolicy-M |
|---|---|---|---|---|---|---|
| | Subset (CoT) | All (Non-CoT) | Subset (CoT) | All (Non-CoT) | Acc (N=6) | Acc (N=6) |
| SFT | ✓ | | 2.5K | 20K | 17.80 | 14.30 |
| SFT | ✓ | ✓ | 2.5K | 20K | 22.25 | 14.15 |
| SFT + GRPO | ✓ | | 2.5K | 20K | 34.10 | 54.00 |
| SFT + GRPO | ✓ | ✓ | 2.5K | 20K | 47.05 | 64.50 |
| TriMPI w/ GRPO | ✓ | | 2.5K | 20K | 39.45 | 74.55 |
| TriMPI w/ GRPO | | ✓ | 2.5K | 20K | 55.90 | 80.80 |

that represent new visual concepts, enabling models to generate personalized responses. However, the supported concepts are typically limited to one or a few personal photos, which is far less complex than the policies required for multimodal conversational agents.

# M    ERROR EXAMPLES

We present the figures supporting the discussion in §5.6.

Figures 11, 12, and 13 show qualitative examples of the common error types.

Figure 14 shows the branching error analysis compared with baselines.

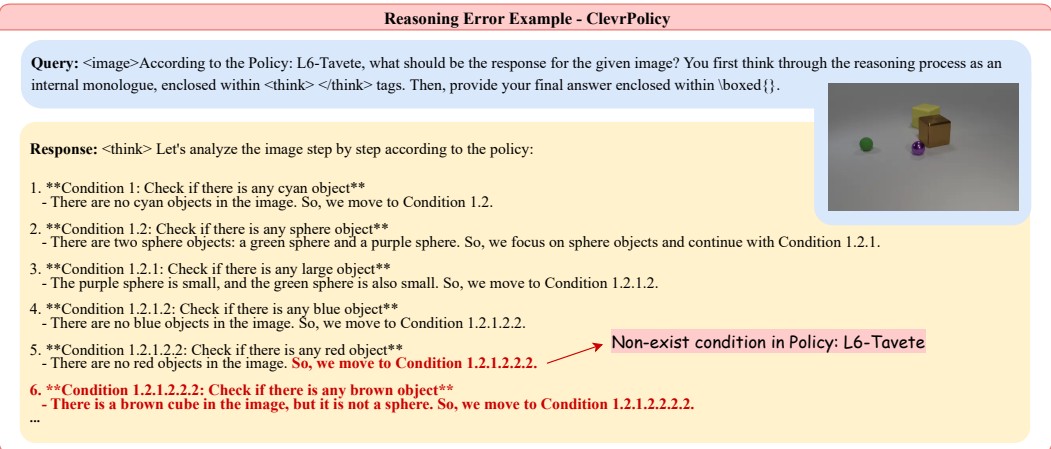

Figure 11: **Reasoning error example on ClevrPolicy.** The model branches to a non-existent Condition for the current policy.

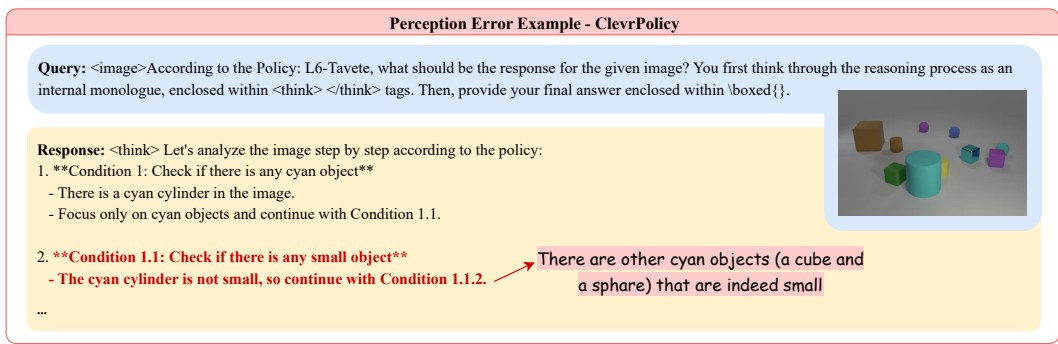

Figure 12: **Perception error example on ClevrPolicy.** The model incorrectly perceives that there are no small cyan objects.

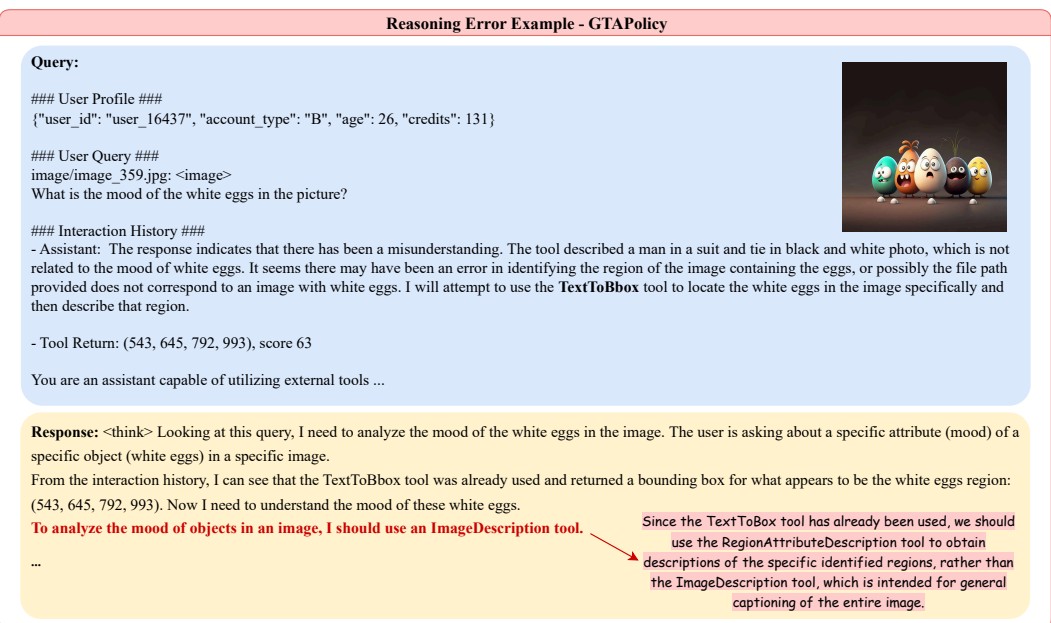

Figure 13: **Reasoning error example on GTAPolicy.** After correctly using "TextToBbox" to localize the region of interest for the query, the model should invoke "RegionAttributeDescription" to describe that specific region, but instead erroneously calls "ImageDescription", which captions the entire image and disregards the prior tool output.

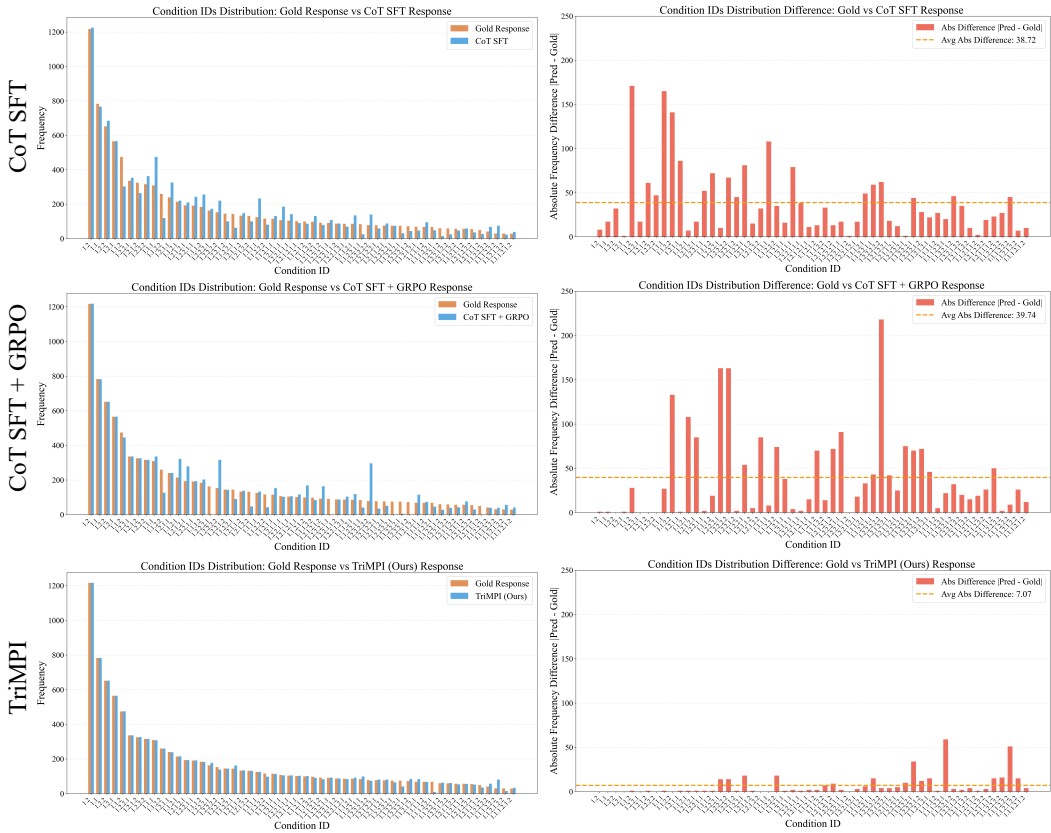

Figure 14: **Branching error analysis on ClevrPolicy.** We compare the distribution of referenced Condition IDs (e.g., "1.1") in model-generated responses against those in a synthesized Gold CoT response. A distribution closer to the Gold CoT distribution (left) indicates better alignment with the original policy and fewer hallucinations, while a smaller difference (right) reflects better overall performance.

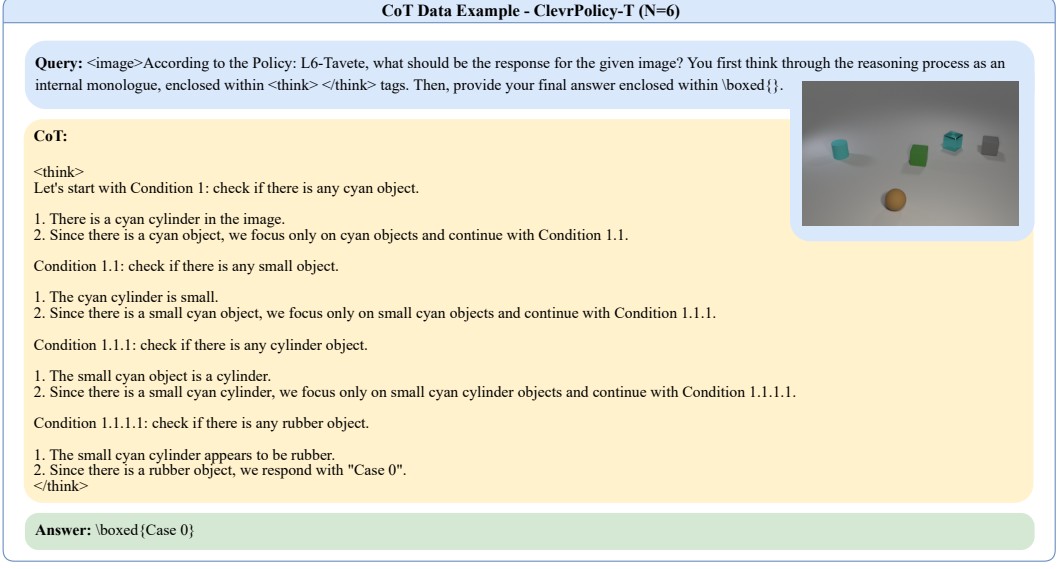

Figure 15: **CoT data example (ClevrPolicy-T N=6).**

---

**CoT Data Example - ClevrPolicy-M (N=6)**

**Query:** <image>According to the Policy: L6-Kagute, what should be the response for the given image? You first think through the reasoning process as an internal monologue, enclosed within <think> </think> tags. Then, provide your final answer enclosed within \boxed{}.

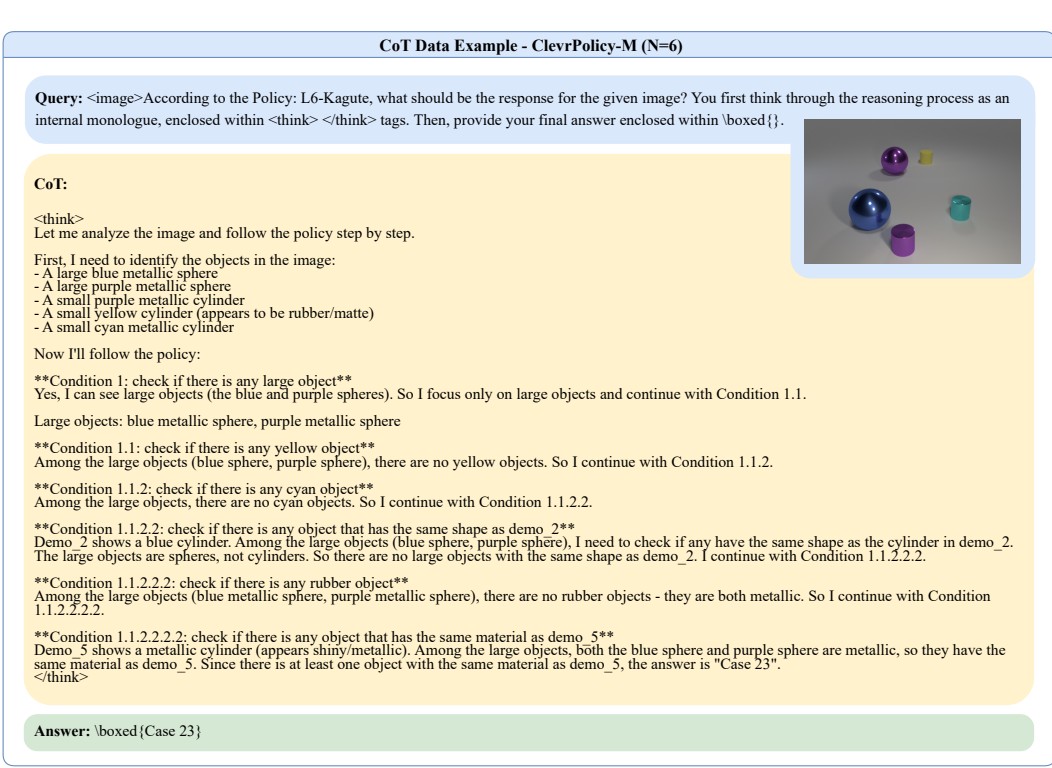

**CoT:**

<think>
Let me analyze the image and follow the policy step by step.

First, I need to identify the objects in the image:
- A large blue metallic sphere
- A large purple metallic sphere
- A small purple metallic cylinder
- A small yellow cylinder (appears to be rubber/matte)
- A small cyan metallic cylinder

Now I'll follow the policy:

**Condition 1: check if there is any large object**
Yes, I can see large objects (the blue and purple spheres). So I focus only on large objects and continue with Condition 1.1.

Large objects: blue metallic sphere, purple metallic sphere

**Condition 1.1: check if there is any yellow object**
Among the large objects (blue sphere, purple sphere), there are no yellow objects. So I continue with Condition 1.1.2.

**Condition 1.1.2: check if there is any cyan object**
Among the large objects, there are no cyan objects. So I continue with Condition 1.1.2.2.

**Condition 1.1.2.2: check if there is any object that has the same shape as demo_2**
Demo_2 shows a blue cylinder. Among the large objects (blue sphere, purple sphere), I need to check if any have the same shape as the cylinder in demo_2. The large objects are spheres, not cylinders. So there are no large objects with the same shape as demo_2. I continue with Condition 1.1.2.2.2.

**Condition 1.1.2.2.2: check if there is any rubber object**
Among the large objects (blue metallic sphere, purple metallic sphere), there are no rubber objects - they are both metallic. So I continue with Condition 1.1.2.2.2.2.

**Condition 1.1.2.2.2.2: check if there is any object that has the same material as demo_5**
Demo_5 shows a metallic cylinder (appears shiny/metallic). Among the large objects, both the blue sphere and purple sphere are metallic, so they have the same material as demo_5. Since there is at least one object with the same material as demo_5, the answer is "Case 23".
</think>

**Answer:** \boxed{Case 23}

---

Figure 16: **CoT data example (ClevrPolicy-M N=6).**

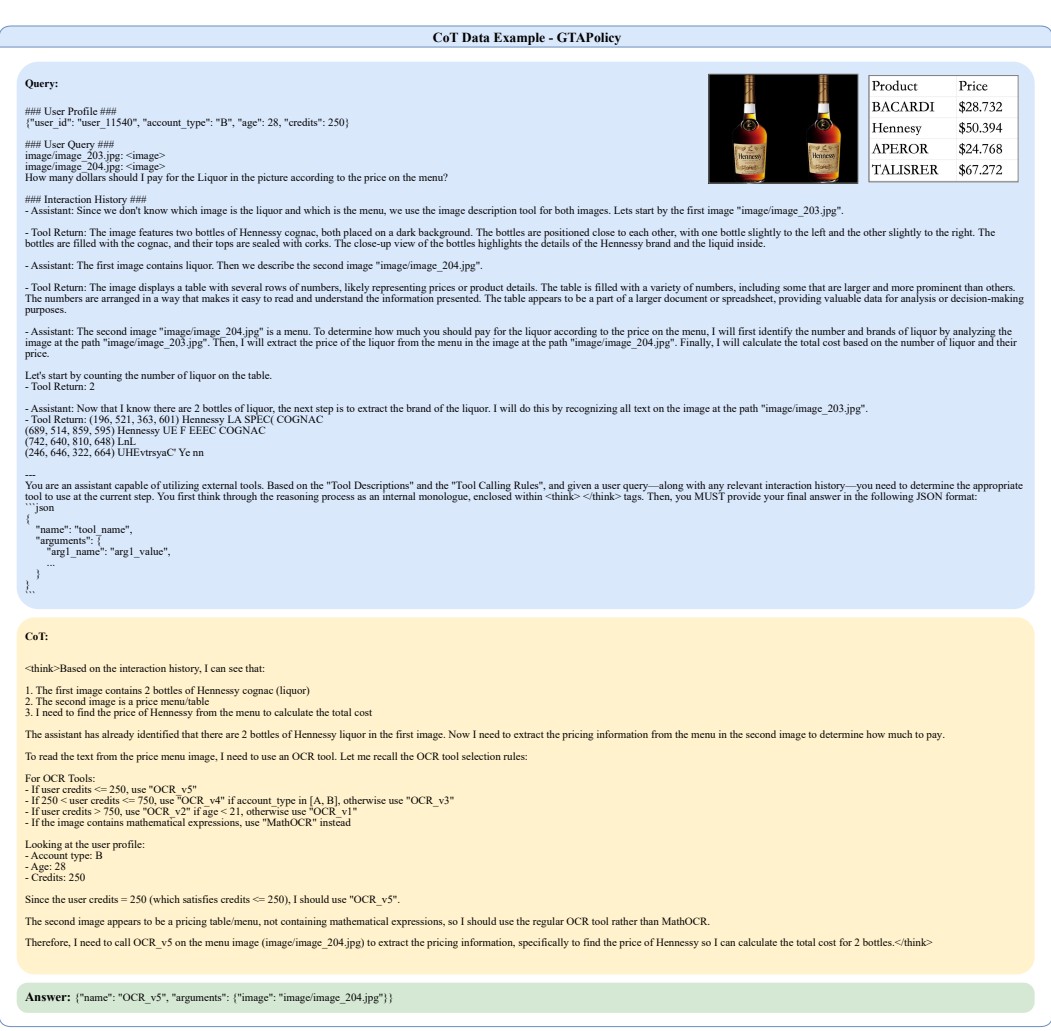

Figure 17: **CoT data example (GTAPolicy).**

**Full Policy Example - ClevrPolicy-T (N=6)**

The following policy provides a decision-making framework for processing a visual input. You should always start with checking the highest-level condition (i.e., Condition 1) and follow the instructions to produce a final response. Lower-level conditions should be evaluated on the subset of objects that satisfy the higher-level conditions, if applicable. The policy is structured as follows:

--- Policy: L6-Tavete ---
# Condition 1: check if there is any cyan object
- If yes, focus only on cyan objects, and continue with Condition 1.1
- Otherwise, continue with Condition 1.2

## Condition 1.1: check if there is any small object
- If yes, focus only on small cyan objects, and continue with Condition 1.1.1
- Otherwise, continue with Condition 1.1.2

### Condition 1.1.1: check if there is any cylinder object
- If yes, focus only on small cyan cylinder objects, and continue with Condition 1.1.1.1
- Otherwise, continue with Condition 1.1.1.2

#### Condition 1.1.1.1: check if there is any rubber object
- If yes, respond "Case 0"
- Otherwise, continue with Condition 1.1.1.1.2

##### Condition 1.1.1.1.2: check if there is any metal object
- If yes, respond "Case 1"
- Otherwise, respond "Case 2"

#### Condition 1.1.1.2: check if there is any sphere object
- If yes, focus only on small cyan sphere objects, and continue with Condition 1.1.1.2.1
- Otherwise, continue with Condition 1.1.1.2.2

##### Condition 1.1.1.2.1: check if there is any metal object
- If yes, respond "Case 3"
- Otherwise, continue with Condition 1.1.1.2.1.2

###### Condition 1.1.1.2.1.2: check if there is any rubber object
- If yes, respond "Case 4"
- Otherwise, respond "Case 5"

##### Condition 1.1.1.2.2: check if there is any metal object
- If yes, focus only on small cyan metal objects, and continue with Condition 1.1.1.2.2.1
- Otherwise, continue with Condition 1.1.1.2.2.2

###### Condition 1.1.1.2.2.1: check if there is any cube object
- If yes, respond "Case 6"
- Otherwise, respond "Case 7"

###### Condition 1.1.1.2.2.2: check if there is any cube object
- If yes, respond "Case 8"
- Otherwise, respond "Case 9"

### Condition 1.1.2: check if there is any cylinder object
- If yes, focus only on cyan cylinder objects, and continue with Condition 1.1.2.1
- Otherwise, continue with Condition 1.1.2.2

#### Condition 1.1.2.1: check if there is any large object
- If yes, focus only on large cyan cylinder objects, and continue with Condition 1.1.2.1.1
- Otherwise, continue with Condition 1.1.2.1.2

##### Condition 1.1.2.1.1: check if there is any rubber object
- If yes, respond "Case 10"
- Otherwise, continue with Condition 1.1.2.1.1.2

###### Condition 1.1.2.1.1.2: check if there is any metal object
- If yes, respond "Case 11"
- Otherwise, respond "Case 12"

##### Condition 1.1.2.1.2: check if there is any metal object
- If yes, respond "Case 13"
- Otherwise, continue with Condition 1.1.2.1.2.2

###### Condition 1.1.2.1.2.2: check if there is any rubber object
- If yes, respond "Case 14"
- Otherwise, respond "Case 15"

#### Condition 1.1.2.2: check if there is any metal object
- If yes, focus only on cyan metal objects, and continue with Condition 1.1.2.2.1
- Otherwise, continue with Condition 1.1.2.2.2

##### Condition 1.1.2.2.1: check if there is any cube object
- If yes, focus only on cyan metal cube objects, and continue with Condition 1.1.2.2.1.1
- Otherwise, continue with Condition 1.1.2.2.1.2

###### Condition 1.1.2.2.1.1: check if there is any large object
- If yes, respond "Case 16"
- Otherwise, respond "Case 17"

###### Condition 1.1.2.2.1.2: check if there is any sphere object
- If yes, respond "Case 18"
- Otherwise, respond "Case 19"

##### Condition 1.1.2.2.2: check if there is any large object
- If yes, focus only on large cyan objects, and continue with Condition 1.1.2.2.2.1
- Otherwise, continue with Condition 1.1.2.2.2.2

###### Condition 1.1.2.2.2.1: check if there is any cube object
- If yes, respond "Case 20"
- Otherwise, respond "Case 21"

###### Condition 1.1.2.2.2.2: check if there is any cube object
- If yes, respond "Case 22"
- Otherwise, respond "Case 23"

## Condition 1.2: check if there is any sphere object
- If yes, focus only on sphere objects, and continue with Condition 1.2.1
- Otherwise, continue with Condition 1.2.2

### Condition 1.2.1: check if there is any large object
- If yes, focus only on large sphere objects, and continue with Condition 1.2.1.1
- Otherwise, continue with Condition 1.2.1.2

#### Condition 1.2.1.1: check if there is any green object
- If yes, focus only on large green sphere objects, and continue with Condition 1.2.1.1.1
- Otherwise, continue with Condition 1.2.1.1.2

##### Condition 1.2.1.1.1: check if there is any metal object
- If yes, respond "Case 24"
- Otherwise, continue with Condition 1.2.1.1.1.2

###### Condition 1.2.1.1.1.2: check if there is any rubber object
- If yes, respond "Case 25"
- Otherwise, respond "Case 26"

##### Condition 1.2.1.1.2: check if there is any metal object
- If yes, focus only on large metal sphere objects, and continue with Condition 1.2.1.1.2.1
- Otherwise, continue with Condition 1.2.1.1.2.2

###### Condition 1.2.1.1.2.1: check if there is any red object
- If yes, respond "Case 27"
- Otherwise, respond "Case 28"

###### Condition 1.2.1.1.2.2: check if there is any gray object
- If yes, respond "Case 29"
- Otherwise, respond "Case 30"

#### Condition 1.2.1.2: check if there is any blue object
- If yes, focus only on blue sphere objects, and continue with Condition 1.2.1.2.1
- Otherwise, continue with Condition 1.2.1.2.2

##### Condition 1.2.1.2.1: check if there is any small object
- If yes, focus only on small blue sphere objects, and continue with Condition 1.2.1.2.1.1
- Otherwise, continue with Condition 1.2.1.2.1.2

###### Condition 1.2.1.2.1.1: check if there is any rubber object
- If yes, respond "Case 31"
- Otherwise, respond "Case 32"

###### Condition 1.2.1.2.1.2: check if there is any rubber object
- If yes, respond "Case 33"
- Otherwise, respond "Case 34"

##### Condition 1.2.1.2.2: check if there is any red object
- If yes, focus only on red sphere objects, and continue with Condition 1.2.1.2.2.1
- Otherwise, continue with Condition 1.2.1.2.2.2

###### Condition 1.2.1.2.2.1: check if there is any small object
- If yes, respond "Case 35"
- Otherwise, respond "Case 36"

###### Condition 1.2.1.2.2.2: check if there is any brown object
- If yes, respond "Case 37"
- Otherwise, respond "Case 38"

### Condition 1.2.2: check if there is any rubber object
- If yes, focus only on rubber objects, and continue with Condition 1.2.2.1
- Otherwise, continue with Condition 1.2.2.2

#### Condition 1.2.2.1: check if there is any gray object
- If yes, focus only on gray rubber objects, and continue with Condition 1.2.2.1.1
- Otherwise, continue with Condition 1.2.2.1.2

##### Condition 1.2.2.1.1: check if there is any small object
- If yes, focus only on small gray rubber objects, and continue with Condition 1.2.2.1.1.1
- Otherwise, continue with Condition 1.2.2.1.1.2

###### Condition 1.2.2.1.1.1: check if there is any cylinder object
- If yes, respond "Case 39"
- Otherwise, respond "Case 40"

###### Condition 1.2.2.1.1.2: check if there is any large object
- If yes, respond "Case 41"
- Otherwise, respond "Case 42"

##### Condition 1.2.2.1.2: check if there is any cube object
- If yes, focus only on rubber cube objects, and continue with Condition 1.2.2.1.2.1
- Otherwise, continue with Condition 1.2.2.1.2.2

###### Condition 1.2.2.1.2.1: check if there is any large object
- If yes, respond "Case 43"
- Otherwise, respond "Case 44"

###### Condition 1.2.2.1.2.2: check if there is any green object
- If yes, respond "Case 45"
- Otherwise, respond "Case 46"

#### Condition 1.2.2.2: check if there is any gray object
- If yes, focus only on gray objects, and continue with Condition 1.2.2.2.1
- Otherwise, continue with Condition 1.2.2.2.2

##### Condition 1.2.2.2.1: check if there is any cylinder object
- If yes, focus only on gray cylinder objects, and continue with Condition 1.2.2.2.1.1
- Otherwise, continue with Condition 1.2.2.2.1.2

###### Condition 1.2.2.2.1.1: check if there is any metal object
- If yes, respond "Case 47"
- Otherwise, respond "Case 48"

###### Condition 1.2.2.2.1.2: check if there is any cube object
- If yes, respond "Case 49"
- Otherwise, respond "Case 50"

##### Condition 1.2.2.2.2: check if there is any large object
- If yes, focus only on large objects, and continue with Condition 1.2.2.2.2.1
- Otherwise, continue with Condition 1.2.2.2.2.2

###### Condition 1.2.2.2.2.1: check if there is any yellow object
- If yes, respond "Case 51"
- Otherwise, respond "Case 52"

###### Condition 1.2.2.2.2.2: check if there is any green object
- If yes, respond "Case 53"
- Otherwise, respond "Case 54"

Figure 18: **Full policy example (ClevrPolicy-T N=6).**

**Full Policy Example - ClevrPolicy-M (N=6)**

The following policy provides a decision-making framework for processing a visual input. You should always start with checking the highest-level condition (i.e., Condition 1) and follow the instructions to produce a final response. Lower-level conditions should be evaluated on the subset of objects that satisfy the higher-level conditions, if applicable. The policy is structured as follows:

--- Policy: L6-Kagute ---
\# Condition 1: check if there is any large object
- If yes, focus only on large objects, and continue with Condition 1.1
- Otherwise, continue with Condition 1.2

\## Condition 1.1: check if there is any yellow object
- If yes, focus only on large yellow objects, and continue with Condition 1.1.1
- Otherwise, continue with Condition 1.1.2

\### Condition 1.1.1: check if there is any cube object
- If yes, focus only on large yellow cube objects, and continue with Condition 1.1.1.1
- Otherwise, continue with Condition 1.1.1.2

\#### Condition 1.1.1.1: check if there is any rubber object
- If yes, respond "Case 0"
- Otherwise, continue with Condition 1.1.1.1.2

\##### Condition 1.1.1.1.2: check if there is any metal object
- If yes, respond "Case 1"
- Otherwise, respond "Case 2"

\#### Condition 1.1.1.2: check if there is any metal object
- If yes, focus only on large yellow metal objects, and continue with Condition 1.1.1.2.1
- Otherwise, continue with Condition 1.1.1.2.2

\##### Condition 1.1.1.2.1: check if there is any sphere object
- If yes, respond "Case 3"
- Otherwise, continue with Condition 1.1.1.2.1.2

\###### Condition 1.1.1.2.1.2: check if there is any cylinder object
- If yes, respond "Case 4"
- Otherwise, respond "Case 5"

\##### Condition 1.1.1.2.2: check if there is any cylinder object
- If yes, focus only on large yellow cylinder objects, and continue with Condition 1.1.1.2.2.1
- Otherwise, continue with Condition 1.1.1.2.2.2

\###### Condition 1.1.1.2.2.1: check if there is any rubber object
- If yes, respond "Case 6"
- Otherwise, respond "Case 7"

\###### Condition 1.1.1.2.2.2: check if there is any rubber object
- If yes, respond "Case 8"
- Otherwise, respond "Case 9"

\### Condition 1.1.2: check if there is any cyan object
- If yes, focus only on large cyan objects, and continue with Condition 1.1.2.1
- Otherwise, continue with Condition 1.1.2.2

\#### Condition 1.1.2.1: check if there is any cylinder object
- If yes, focus only on large cyan cylinder objects, and continue with Condition 1.1.2.1.1
- Otherwise, continue with Condition 1.1.2.1.2

\##### Condition 1.1.2.1.1: check if there is any metal object
- If yes, respond "Case 10"
- Otherwise, continue with Condition 1.1.2.1.1.2

\###### Condition 1.1.2.1.1.2: check if there is any rubber object
- If yes, respond "Case 11"
- Otherwise, respond "Case 12"

\##### Condition 1.1.2.1.2: check if there is any object that has the same material as: demo_0<image>
- If yes, focus only on large cyan objects with the same material as demo_0, and continue with Condition 1.1.2.1.2.1
- Otherwise, continue with Condition 1.1.2.1.2.2

\###### Condition 1.1.2.1.2.1: check if there is any object that has the same shape as: demo_1<image>
- If yes, respond "Case 13"
- Otherwise, respond "Case 14"

\###### Condition 1.1.2.1.2.2: check if there is any metal object
- If yes, respond "Case 15"
- Otherwise, respond "Case 16"

\#### Condition 1.1.2.2: check if there is any object that has the same shape as: demo_2<image>
- If yes, focus only on large objects with the same shape as demo_2, and continue with Condition 1.1.2.2.1
- Otherwise, continue with Condition 1.1.2.2.2

\##### Condition 1.1.2.2.1: check if there is any object that has the same color as: demo_3<image>
- If yes, focus only on large objects with the same color as demo_3, the same shape as demo_2, and continue with Condition 1.1.2.2.1.1
- Otherwise, continue with Condition 1.1.2.2.1.2

\###### Condition 1.1.2.2.1.1: check if there is any metal object
- If yes, respond "Case 17"
- Otherwise, respond "Case 18"

\###### Condition 1.1.2.2.1.2: check if there is any object that has the same material as: demo_4<image>
- If yes, respond "Case 19"
- Otherwise, respond "Case 20"

\##### Condition 1.1.2.2.2: check if there is any rubber object
- If yes, focus only on large rubber objects, and continue with Condition 1.1.2.2.2.1
- Otherwise, continue with Condition 1.1.2.2.2.2

\###### Condition 1.1.2.2.2.1: check if there is any cube object
- If yes, respond "Case 21"
- Otherwise, respond "Case 22"

\###### Condition 1.1.2.2.2.2: check if there is any object that has the same material as: demo_5<image>
- If yes, respond "Case 23"
- Otherwise, respond "Case 24"

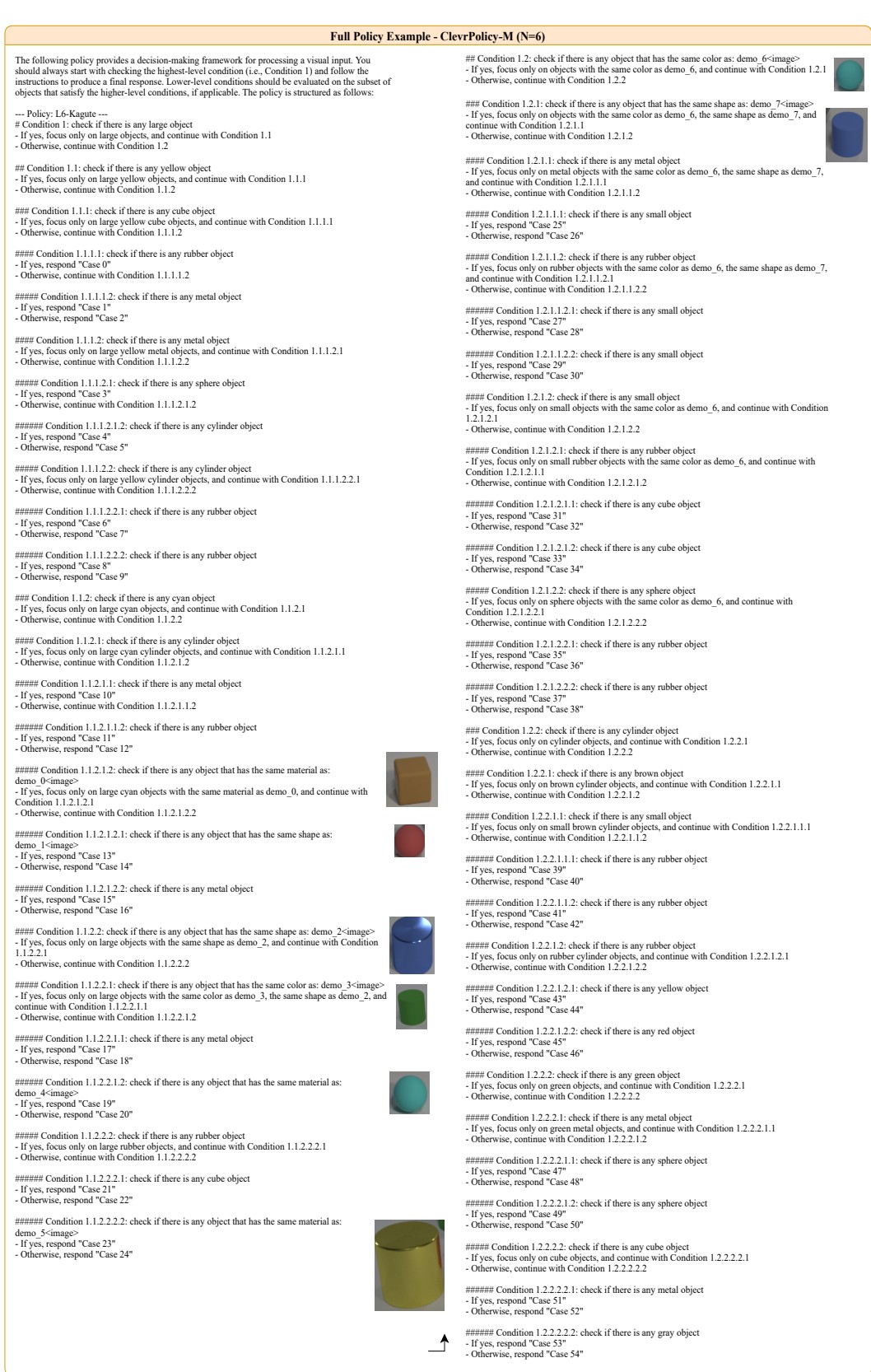

\## Condition 1.2: check if there is any object that has the same color as: demo_6<image>
- If yes, focus only on objects with the same color as demo_6, and continue with Condition 1.2.1
- Otherwise, continue with Condition 1.2.2

\### Condition 1.2.1: check if there is any object that has the same shape as: demo_7<image>
- If yes, focus only on objects with the same color as demo_6, the same shape as demo_7, and continue with Condition 1.2.1.1
- Otherwise, continue with Condition 1.2.1.2

\#### Condition 1.2.1.1: check if there is any metal object
- If yes, focus only on metal objects with the same color as demo_6, the same shape as demo_7, and continue with Condition 1.2.1.1.1
- Otherwise, continue with Condition 1.2.1.1.2

\##### Condition 1.2.1.1.1: check if there is any small object
- If yes, respond "Case 25"
- Otherwise, respond "Case 26"

\##### Condition 1.2.1.1.2: check if there is any rubber object
- If yes, focus only on rubber objects with the same color as demo_6, the same shape as demo_7, and continue with Condition 1.2.1.1.2.1
- Otherwise, continue with Condition 1.2.1.1.2.2

\###### Condition 1.2.1.1.2.1: check if there is any small object
- If yes, respond "Case 27"
- Otherwise, respond "Case 28"

\###### Condition 1.2.1.1.2.2: check if there is any small object
- If yes, respond "Case 29"
- Otherwise, respond "Case 30"

\#### Condition 1.2.1.2: check if there is any small object
- If yes, focus only on small objects with the same color as demo_6, and continue with Condition 1.2.1.2.1
- Otherwise, continue with Condition 1.2.1.2.2

\##### Condition 1.2.1.2.1: check if there is any rubber object
- If yes, focus only on small rubber objects with the same color as demo_6, and continue with Condition 1.2.1.2.1.1
- Otherwise, continue with Condition 1.2.1.2.1.2

\###### Condition 1.2.1.2.1.1: check if there is any cube object
- If yes, respond "Case 31"
- Otherwise, respond "Case 32"

\###### Condition 1.2.1.2.1.2: check if there is any cube object
- If yes, respond "Case 33"
- Otherwise, respond "Case 34"

\##### Condition 1.2.1.2.2: check if there is any sphere object
- If yes, focus only on sphere objects with the same color as demo_6, and continue with Condition 1.2.1.2.2.1
- Otherwise, continue with Condition 1.2.1.2.2.2

\###### Condition 1.2.1.2.2.1: check if there is any rubber object
- If yes, respond "Case 35"
- Otherwise, respond "Case 36"

\###### Condition 1.2.1.2.2.2: check if there is any rubber object
- If yes, respond "Case 37"
- Otherwise, respond "Case 38"

\### Condition 1.2.2: check if there is any cylinder object
- If yes, focus only on cylinder objects, and continue with Condition 1.2.2.1
- Otherwise, continue with Condition 1.2.2.2

\#### Condition 1.2.2.1: check if there is any brown object
- If yes, focus only on brown cylinder objects, and continue with Condition 1.2.2.1.1
- Otherwise, continue with Condition 1.2.2.1.2

\##### Condition 1.2.2.1.1: check if there is any small object
- If yes, focus only on small brown cylinder objects, and continue with Condition 1.2.2.1.1.1
- Otherwise, continue with Condition 1.2.2.1.1.2

\###### Condition 1.2.2.1.1.1: check if there is any rubber object
- If yes, respond "Case 39"
- Otherwise, respond "Case 40"

\###### Condition 1.2.2.1.1.2: check if there is any rubber object
- If yes, respond "Case 41"
- Otherwise, respond "Case 42"

\##### Condition 1.2.2.1.2: check if there is any rubber object
- If yes, focus only on rubber cylinder objects, and continue with Condition 1.2.2.1.2.1
- Otherwise, continue with Condition 1.2.2.1.2.2

\###### Condition 1.2.2.1.2.1: check if there is any yellow object
- If yes, respond "Case 43"
- Otherwise, respond "Case 44"

\###### Condition 1.2.2.1.2.2: check if there is any red object
- If yes, respond "Case 45"
- Otherwise, respond "Case 46"

\#### Condition 1.2.2.2: check if there is any green object
- If yes, focus only on green objects, and continue with Condition 1.2.2.2.1
- Otherwise, continue with Condition 1.2.2.2.2

\##### Condition 1.2.2.2.1: check if there is any metal object
- If yes, focus only on green metal objects, and continue with Condition 1.2.2.2.1.1
- Otherwise, continue with Condition 1.2.2.2.1.2

\###### Condition 1.2.2.2.1.1: check if there is any sphere object
- If yes, respond "Case 47"
- Otherwise, respond "Case 48"

\###### Condition 1.2.2.2.1.2: check if there is any sphere object
- If yes, respond "Case 49"
- Otherwise, respond "Case 50"

\##### Condition 1.2.2.2.2: check if there is any cube object
- If yes, focus only on cube objects, and continue with Condition 1.2.2.2.2.1
- Otherwise, continue with Condition 1.2.2.2.2.2

\###### Condition 1.2.2.2.2.1: check if there is any metal object
- If yes, respond "Case 51"
- Otherwise, respond "Case 52"

\###### Condition 1.2.2.2.2.2: check if there is any gray object
- If yes, respond "Case 53"
- Otherwise, respond "Case 54"

Figure 19: **Full policy example (ClevrPolicy-M N=6).**

---

**Full Policy Example - GTAPolicy**

### Tool Descriptions ###
Type: Calculator
Name Versions: Calculator
Description: A calculator tool. The input must be a single Python expression and you cannot import packages. You can use functions in the `math` package without import.
Arguments:[{"name": "expression", "type": "text", "description": null, "required": true, "default": null}]

Type: OCR
Name Versions: OCR_v1, OCR_v2, OCR_v3, OCR_v4, OCR_v5
Description: This tool can recognize all text on the input image.
Arguments:[{"name": "image", "type": "image", "description": null, "required": true, "default": null}]

Type: CountGivenObject
Name Versions: CountGivenObject_v1, CountGivenObject_v2, CountGivenObject_v3, CountGivenObject_v4
Description: The tool can count the number of a certain object in the image.
Arguments:[{"name": "image", "type": "image", "description": null, "required": true, "default": null}, {"name": "text", "type": "text", "description": "The object description in English.", "required": true, "default": null}]

Type: ImageDescription
Name Versions: ImageDescription_v1, ImageDescription_v2, ImageDescription_v3, ImageDescription_v4, ImageDescription_v5
Description: A useful tool that returns a brief description of the input image.
Arguments:[{"name": "image", "type": "image", "description": null, "required": true, "default": null}]

Type: GoogleSearch
Name Versions: GoogleSearch_v1, GoogleSearch_v2, GoogleSearch_v3
Description: The tool can search the input query text from Google and return the related results.
Arguments:[{"name": "query", "type": "text", "description": null, "required": true, "default": null}, {"name": "k", "type": "int", "description": "Select the first k results", "required": false, "default": 10}]

Type: RegionAttributeDescription
Name Versions: RegionAttributeDescription_v1, RegionAttributeDescription_v2
Description: Describe the attribute of a region of the input image.
Arguments:[{"name": "image", "type": "image", "description": null, "required": true, "default": null}, {"name": "bbox", "type": "text", "description": "The bbox coordinate in the format of `(x1, y1, x2, y2)`", "required": true, "default": null}, {"name": "attribute", "type": "text", "description": "The attribute to describe", "required": true, "default": null}]

Type: TextToBbox
Name Versions: TextToBbox
Description: The tool can detect the object location according to description.
Arguments:[{"name": "image", "type": "image", "description": null, "required": true, "default": null}, {"name": "text", "type": "text", "description": "The object description in English.", "required": true, "default": null}, {"name": "top1", "type": "bool", "description": "If true, return the object with highest score. If false, return all detected objects.", "required": false, "default": true}]

Type: Plot
Name Versions: Plot
Description: This tool can execute Python code to plot diagrams. The code should include a function named 'solution'. The function should return the matplotlib figure directly. Avoid printing the answer. The code instance format is as follows:

```python
# import packages
import matplotlib.pyplot as plt
def solution():
    # labels and data
    cars = ['AUDI', 'BMW', 'FORD', 'TESLA', 'JAGUAR', 'MERCEDES']
    data = [23, 17, 35, 29, 12, 41]

    # draw diagrams
    figure = plt.figure(figsize=(8, 6))
    plt.pie(data, labels=cars, autopct='%1.1f%%', startangle=140)
    plt.axis('equal')
    plt.title('Car Distribution')
    return figure
```
Arguments:[{"name": "command", "type": "text", "description": "Markdown format Python code", "required": true, "default": null}]

Type: MathOCR
Name Versions: MathOCR
Description: This tool can recognize math expressions from an image and return the latex style expression.
Arguments:[{"name": "image", "type": "image", "description": null, "required": true, "default": null}]

Type: Solver
Name Versions: Solver
Description: This tool can execute Python code to solve math equations. The code should include a function named 'solution'. You should use the `sympy` library in your code to solve the equations. The function should return its answer in str format. Avoid printing the answer. The code instance format is as follows:

```python
# import packages
from sympy import symbols, Eq, solve
def solution():
    # Define symbols
    x, y = symbols('x y')

    # Define equations
    equation1 = Eq(x**2 + y**2, 20)
    equation2 = Eq(x**2 - 5*x*y + 6*y**2, 0)

    # Solve the system of equations
    solutions = solve((equation1, equation2), (x, y), dict=True)

    # Return solutions as strings
    return str(solutions)
```
Arguments:[{"name": "command", "type": "text", "description": "Markdown format Python code", "required": true, "default": null}]

Type: DrawBox
Name Versions: DrawBox
Description: A tool to draw a box on a certain region of the input image.
Arguments:[{"name": "image", "type": "image", "description": null, "required": true, "default": null}, {"name": "bbox", "type": "text", "description": "The bbox coordinate in the format of `(x1, y1, x2, y2)`", "required": true, "default": null}, {"name": "annotation", "type": "text", "description": "The extra annotation text of the bbox", "required": false, "default": null}]

Type: AddText
Name Versions: AddText
Description: A tool to draw a box on a certain region of the input image.
Arguments:[{"name": "image", "type": "image", "description": null, "required": true, "default": null}, {"name": "text", "type": "text", "description": null, "required": true, "default": null}, {"name": "position", "type": "text", "description": "The left-bottom corner coordinate in the format of `(x, y)`, or a combination of [\"l\"(left), \"m\"(middle), \"r\"(right)] and [\"t\"(top), \"m\"(middle), \"b\"(bottom)] like \"mt\" for middle-top", "required": true, "default": null}, {"name": "color", "type": "text", "description": null, "required": false, "default": "red"}]

Type: TextToImage
Name Versions: TextToImage_v1, TextToImage_v2, TextToImage_v3
Description: This tool can generate an image according to the input text.
Arguments:[{"name": "keywords", "type": "text", "description": "A series of keywords separated by comma.", "required": true, "default": null}]

Type: ImageStylization
Name Versions: ImageStylization_v1, ImageStylization_v2, ImageStylization_v3
Description: This tool can modify the input image according to the input instruction. Here are some example instructions: "turn him into cyborg", "add fireworks to the sky", "make his jacket out of leather".
Arguments:[{"name": "image", "type": "image", "description": null, "required": true, "default": null}, {"name": "instruction", "type": "text", "description": null, "required": true, "default": null}]

### Tool Calling Rules ###
[ImageDescription Tools]:
- If user account_type is A, always use "ImageDescription_v5".
- If user account_type is B or C, use "ImageDescription_v4" if user credits > 500, otherwise use "ImageDescription_v3".
- If user account_type is D, use "ImageDescription_v2" if user credits > 500, otherwise use "ImageDescription_v1".

[OCR Tools]:
- If user credits <= 250, use "OCR_v5".
- If 250 < user credits <= 750, use "OCR_v4" if account_type in [A, B], otherwise use "OCR_v3".
- If user credits > 750, use "OCR_v2" if age < 21, otherwise use "OCR_v1".
- If the image contains mathematical expressions, use "MathOCR" instead of the "OCR_*" tools.

[GoogleSearch Tools]:
- If user age < 21, use "GoogleSearch_v1".
- If user age >= 21, use "GoogleSearch_v2" if user credits < 500, otherwise use "GoogleSearch_v3".

[TextToImage Tools]:
- If user account_type in [A, B, C], always use "TextToImage_v1".
- If user account_type is D, use "TextToImage_v2" if user credits > 500, otherwise use "TextToImage_v3".

[RegionAttributeDescription Tools]:
- If user account_type is A or B, always use "RegionAttributeDescription_v2".
- If user account_type is C or D, always use "RegionAttributeDescription_v1".

[CountGivenObject Tools]:
- If user credits < 250, use "CountGivenObject_v1".
- If 250 <= user credits < 500, use "CountGivenObject_v2".
- If 500 <= user credits < 750, use "CountGivenObject_v3".
- If user credits >= 750, use "CountGivenObject_v4".

[ImageStylization Tools]:
- If user age < 21, use "ImageStylization_v1".
- If 21 <= user age < 60, use "ImageStylization_v2".
- If user age >= 60, use "ImageStylization_v3".

[Additional Tips]:
- When there are multiple images provided. Always call one of the "ImageDescription_*" tools on each of the images if you haven't done so.
- Before calling one of the "RegionAttributeDescription_*" tools you should first get the bbox of the region via "TextToBbox", the format to the "bbox" argument is (x1, y1, x2, y2).
- When providing the argument values as images, you should use the exact image name such as "image/image_1.jpg".
- When using "GoogleSearch_*" tools, use "k=1" if the question is not open-ended, otherwise use "k=4".

Figure 20: **Full policy example (GTAPolicy).**

