# OpenReview forum: "Multimodal Policy Internalization for Conversational Agents"
_ICLR.cc/2026/Conference — ICLR 2026 Poster_

### Official Review · Reviewer_2FLG · 2025-10-28

**Soundness:** 3
**Presentation:** 4
**Contribution:** 3
**Rating:** 6
**Confidence:** 3

**Summary:**

This work addresses the challenge of Multimodal Policy Internalization. Policy internalization is a known problem in LLMs, which involves learning policy information through model parameters, instead of providing it in context during inference. This work highlights that this in-context policy can significantly increase inference costs, more so for simpler user queries. This work concretely explores these issues with Multimodal models and proposes a benchmark and a training paradigm for multimodal policy internalization. Overall, on the proposed benchmark, with their framework TriMPI, the multimodal models are much better at following policy and accomplishing a task, without requiring the policy in context during inference.

**Strengths:**

- Firstly, I think the paper was well written, and most of the content was very clearly understandable. Great job on that!
- The problem of multimodal policy internalization is a well-motivated one, especially in scenarios like legal or financial applications, where agentics systems might need to reference large documents with text and images to make decisions.
- The idea of policy rollout, to include policy-aware algorithms in the response space, is a clever one and seems to have good empirical benefits.
- The authors have ablated various stages of their pipeline, CPT, RL finetuning, and the role of policy rollout.
- Additional analysis on computational efficiency and evaluating policy referral is interesting and should provide useful advice for future practitioners when adopting the ideas from this work.

**Weaknesses:**

One key question I have about this work is that, although it is geared towards multimodal models, it seems that most ideas are in some way bootstrapped on ideas in the language domain, and they have, after some tuning, shown benefits.
- This is not necessarily a bad thing, but were there any explorations as to how using policy images can benefit policy internalization?
- The lack of discussion on the vision side of things is my only major complaint about this work.

**Questions:**

- L304, it's claimed that the policy remains insufficiently grounded in the policy. Was this a qualitative observation? Is there a way it can be measured?
- Just as an additional baseline, what if the model did have access to the prompt at inference time?
  - Does MPI still work better?
  - Or does having the prompt, essentially, make all the baselines and MPI the same in terms of performance?
  - This could be a good tradeoff to understand when the cost of MPI tuning may be higher than just putting the policy in the prompt. Note that this does not take any away from the contribution of the work.
- At what point does the in-context policy tradeoff wrt the user query become negligible?
  - Specifically, are there scenarios where the user queries may be large enough that the policy tokens are negligible with respect to them?
- In the policy rollout, the way I understand it, is that the RL algorithm now has more policy-aware options, but a question comes to mind, which might be naive:
  - How are these policy-aware responses affecting the rollouts? Is it because they’re more correct, hence GRPO will force the Policy model to produce more similar trajectories?
- Is RAG an alternative? One piece of related work that I get curious about when thinking of the problem of multimodal policy internalization is what if the LLM could leverage a RAG-like system to retrieve necessary policy-based information.
  - Maybe this is a different way of implementing MPI, but this can maybe reduce the amount of in-context information needed during inference, by only retrieving the relevant one. I would be curious to hear the author’s thoughts on this one.
- Layers in CLEVR policy: are these referring to the layers in the decision tree? Is that how the complexity of tasks is defined?
- Minor: There are often references to figures and tables in the supplement; perhaps it might become easier to parse if it were stated, like Table 5 in the Supplement, etc.

---

> ### Author Response · Authors · 2025-11-21
>
> **We thank reviewer 2FLG for the detailed and insightful comments. We appreciate the reviewer’s acknowledgement of our strengths and for contributing to a positive and professional discussion. We address the remaining questions and suggestions below, and we have reflected these changes in the updated manuscript.**
>
>
> # W1: Lack of discussion on vision-specific aspects
>
> ## W1.1:
> First, from a method perspective, the reviewer’s assessment is accurate: our proposed approach is not limited to the multimodal domain and can, in fact, be directly applied to policy internalization in text-only settings. As the reviewer noted, we consider this generality to be an advantage. However, echoing our discussion in the limitations section, we also acknowledge that there remains substantial room for improvement in handling visual content within the policy itself. Currently, we employ a simple visual-masking strategy during the VM-CPT stage. Although this design works empirically, we believe that more sophisticated techniques are necessary for more complex visual content.
>
> One potential direction we are considering is incorporating latent visual tokens [1] to better represent visual information during continual pretraining, while still remaining consistent with the overall TriMPI framework. We leave this as an interesting direction for future work.
>
> ## W1.2:
> Second, from a task perspective, we would like to further elaborate on the unique benefits of including images as part of the policy. To make this concrete, we provide several real-world examples. The key motivation is that certain attributes or instructions are difficult to express precisely using natural language alone, yet can be conveyed much more clearly through visual modality. For instance:
> - (1) Personalization-related knowledge: a photo of my pet or family member’s face, where the appearance is hard to describe in language.
> - (2) Visual demonstrations: demo images showing features that are difficult to express in text, such as defects in manufacturing applications or plant illnesses in agricultural settings. The multimodal content in ClevrPolicy-M can be viewed as a simplified attempt to model this application.
> - (3) Visually grounded guidelines: manuals for an app or product where bounding boxes are used to localize buttons, since describing such spatial coordinates purely in text is highly challenging for models to interpret.
>
> ### Reference for W1:
> [1] Li, Bangzheng, et al. "Latent visual reasoning." arXiv preprint arXiv:2509.24251 (2025).
>
>
>
> # Q1: Further explanation on “RL baselines are still insufficiently grounded in the policy”
> The claim originates from qualitative observations that the SFT + GRPO/DAPO models still produce a substantial number of errors when attempting to faithfully ground their reasoning in the rules specified by the original policy. For example, in ClevrPolicy, a typical failure mode involves the model branching into an incorrect or non-existent Condition section (e.g., “1.2.1.2.2.2” in Policy: L6-Tavete).
>
> To further quantify this observation, we conducted an **additional analysis** in which we compared the distribution of referenced Condition IDs in model-generated responses against those in a synthesized Gold CoT response for the ClevrPolicy dataset. We evaluated SFT, SFT + GRPO, and TriMPI models, and the corresponding plots are provided in  **Figure 14 (in Appendix) in the updated manuscript**.
>
> Our results show that GRPO does not meaningfully reduce the discrepancy in Condition ID distribution relative to the SFT baseline (Average Absolute Difference: 38.72 vs. 39.74), indicating that GRPO alone does not sufficiently improve referral accuracy to the policy. In contrast, TriMPI achieves a substantial reduction in this discrepancy (Average Absolute Difference: 7.07), demonstrating that TriMPI enables much more grounded and policy-aligned reasoning.
>
> # Q2: Additional baseline: add policy in-context during inference
> The reviewer raises an interesting experiment that we believe can further deepen our understanding of how the proposed method works. Following this suggestion, we conduct an additional evaluation in which we **include the policy content in the prompt during inference.**
>
> Here are the results on the three benchmarks (ClevrPolicy-T (N=6), ClevrPolicy-M (N=6), and GTAPolicy), comparing the SFT, SFT+GRPO, and TriMPI methods:
>
> | Model | ClevrPolicy-T | ClevrPolicy-M | GTAPolicy Tool Matching | GTAPolicy Argument | GTAPolicy Overall |
> |-|-|-|-|-|-|
> | Base (Qwen2.5-VL-7B) | 13.15 | 5.65 | 23.58 | 19.44 | 21.51 |
> | SFT | 10.70 | 15.80 | 45.28 | 39.24 | 42.26 |
> | SFT + GRPO | 19.00 | 20.90 | 60.38 | 53.64 | 57.01 |
> | **Ours (TriMPI w/ PoRo-GRPO)** | 35.15 | 69.90 | 70.75 | 62.86 | 66.81 |
>
> We observe that under this setting, TriMPI still demonstrates superior performance compared to the baselines. This further justifies the additional training efforts introduced by TriMPI.

---

> > ### Author Response · Authors · 2025-11-21
> >
> > # Q3: At what point does the in-context policy tradeoff wrt the user query become negligible?
> > Yes, we believe there are certain tasks, such as long-document understanding or long-context coding, where the user query can be comparable to or even longer than the policy content (e.g., 10-50K tokens). In most conversational agentic tasks, our observation is that user queries are typically shorter. We take reference from some public statistics on chatbot usage, such as: https://aws.amazon.com/blogs/machine-learning/demystifying-amazon-bedrock-pricing-for-a-chatbot-assistant/
> >
> > # Q4: Further explanation of the benefit of PolicyRollout
> > One of the main challenges of the GRPO/DAPO baselines is that, without access to the original policy, the model’s exploration is largely ungrounded. As further illustrated in Q1, even after SFT, adherence to the original policy remains weak.
> > We believe the primary benefit of PolicyRollout is that it increases the likelihood of sampling meaningful, policy-grounded response trajectories, thereby yielding more informative group advantages and better RL learning signals.
> >
> > # Q5: Discussion on RAG alternatives
> > We agree with the reviewer that RAG can be a strong alternative, or even a complementary component, to the MPI framework.
> >
> > The advantage of RAG is that it is less prone to overfitting and better preserves the model’s original capabilities. On the other hand, it may introduce additional noise due to imperfect retrieval. Retrieval over complex policies is itself a challenging and interesting problem. For example, to answer a single query, the model may need to retrieve multiple snippets or sections from different parts of the policy; in such cases, a typical Top-K (with a fixed K) retrieval based on string similarity may introduce noise in some cases but omit key information in others.
> >
> > Alignment-based methods such as TriMPI directly generate policy-relevant content, which provides more flexibility in these scenarios; however, this comes with the risk of hallucination.
> >
> > Therefore, we believe an interesting direction for future work is to combine the strengths of both approaches: ideally, a model that internalizes the high-level structure of the policy through internalization while leveraging dynamic retrieval to obtain low-level details when needed.
> >
> > # Q6: Clarifications on the “layers” in ClevrPolicy
> > Yes, the layers refer to the number of layers in the binary decision tree and directly reflect its complexity: deeper trees correspond to more branching before reaching the response node, and thus represent more complex policies. We will make this clearer in the updated manuscript.
> >
> >
> > # Q7: Writing improvement: Be specific that a Table is in supplementary
> > We thank the reviewer for the suggestion, we have reflected the changes in the updated manuscript.

---

> > > ### Comment · Reviewer_2FLG · 2025-11-28
> > > **Comments on author response**
> > >
> > > Thank you for the effort.
> > >
> > > One follow-up question,
> > >
> > > For Q2, adding policy in-context during inference, I am a bit confused about the numbers.
> > >
> > > The last row, which is your method, gets, let's say, on GTAPolicy overall, 66.81 with the prompt, but it gets 81.06, without the prompt, as per Table 2 in the draft. I see a similar pattern for other metrics, too.
> > >
> > > Any insight or explanation for this behavior? It seems counterintuitive that adding the prompt in context would degrade the performance, since the premise of this work is that adding prompts in context is compute heavy, hence policy internalization is a useful mechanism.

---

> > > > ### Author Response · Authors · 2025-11-28
> > > >
> > > > We thank reviewer 2FLG for the follow-up question! Yes, we also observed this behavior. We believe it is due to, after internalization, the models are heavily trained on the task: "(no policy) query -> answer", which is inherently different from the task "(policy context) + query -> answer." This mismatch likely also contributes to the degradation we observe when performing policy override using an entirely new policy.
> > > >
> > > > That said, we believe this phenomenon highlights significant room for improving the internalization capability. Ideally, when the original policy is provided in the context, the model should perform equally well or even better, demonstrating a robust and general policy-following ability regardless of whether the policy appears in-context.
> > > >
> > > > We thank the reviewer again for raising this interesting point and contributing to the discussion!

---

### Official Review · Reviewer_DXvn · 2025-11-01

**Soundness:** 3
**Presentation:** 3
**Contribution:** 3
**Rating:** 8
**Confidence:** 3

**Summary:**

The paper tackles the problem of internalizing multimodal policies---complex, reasoning-intensive system prompts that guide agent behavior---into the parameters of large multimodal conversational models to improve efficiency and policy adherence. It introduces TriMPI, a three-stage framework combining Visually-Masked Continual Pretraining (VM-CPT), Chain-of-Thought Supervised Fine-Tuning (CoT-SFT), and PolicyRollout, a policy-aware reinforcement learning stage extending GRPO/DAPO for grounded exploration. Experiments on two proposed benchmarks, ClevrPolicy (synthetic decision-making) and GTAPolicy (tool-use), show significant performance gains over SFT and in-context baselines, while maintaining minimal degradation on general reasoning benchmarks (MMMU-Pro and MMLU-Pro).

**Strengths:**

- The paper is easy to follow, with strong conceptual motivation and clear examples of datasets and methods.
- The paper addresses an important and relevant challenge of conversational agents under long system prompts.
- Extensive experiments, ablations, and analysis (including generalization, efficiency, and catastrophic forgetting) lend strong empirical evidence to claims.

**Weaknesses:**

- It requires multi-stage SFT and RL with explicit SFT data; less straightforward than prompt-based alignment approaches.
- It is evaluated on self-constructed datasets (ClevrPolicy, GTAPolicy) rather than realistic multimodal policy data.
- It is unclear how the benefit of “multimodal policy” over “text-only policy” translates to conversation agent use cases.

**Questions:**

- *L048:* The estimated range of policy prompt lengths (1K–50K tokens) is broad—are there empirical or open-source statistics supporting this claim?
- What are practical scenarios where multimodal policy adherence is critical over text-only policy for conversational agents?
- What’s the distinction with deliberative alignment (Guan et al., 2024) apart from the input modality?
- Writing
    - *L072—073:* The phrase “…, which requires minimal reasoning” can be more clear, example,  “…, which requires minimal reasoning to adhere to the policy.”
    - There’s seems to be inconsistency in Section 4: Line 256 describes two stages while Line 259 and Figure 4 show three. The method description be made more consistent across sections.

---

> ### Author Response · Authors · 2025-11-21
>
> **We thank reviewer DXvn for the detailed and insightful comments. We appreciate the reviewer’s acknowledgement of our strengths and for contributing to a positive and professional discussion. We address the remaining questions and suggestions below, and we have reflected these changes in the updated manuscript.**
>
> # W1: Required SFT and RL, less straightforward than prompt-based alignment
> We acknowledge that MPI requires additional training and data collection. While prompting-based methods are training-free, as shown in the zero-shot prompting results, their expressiveness is quite limited when the policy becomes more complex. Even on state-of-the-art foundation models such as Claude-4, prompting-based policy following struggles to achieve good performance. This motivates us to propose TriMPI, which aims to teach even a small open-source model to follow complex policies, including in low-data regimes such as GTAPolicy.
>
> # W2: Limitation in datasets
> This problem echoes our discussion of limitations in Section 7 regarding the datasets: scaling them up to include larger and more diverse real-world multimodal policies and queries. This is also a primary motivation behind our construction of the two new benchmarks. When designing the ClevrPolicy and GTAPolicy datasets, we deliberately aligned them with realistic scenarios such as “complex multimodal manuals with many sections” and “tool-use rules conditioned on user profiles”.
> We believe that developing a further scaled-up version of these datasets is an important direction for future work and will be essential for advancing research on multimodal policy internalization and alignment.
>
>
> # W3, Q2: Further justification of why “multimodal policy” is needed in conversational agents
> We thank the reviewer for pointing out this lack of clarity. We will explain with real-world examples as follows.
> The main motivation for introducing multimodal content in the policy, such as images and videos, is that some attributes or instructions are difficult to describe in natural language but are much more straightforward in the visual modality. For example:
> - (1) Personalization-related knowledge: a photo of my pet or family member’s face, where the appearance is hard to describe in language.
> - (2) Visual demonstrations: demo images showing features that are difficult to express in text, such as defects in manufacturing applications or plant illnesses in agricultural settings. The multimodal content in ClevrPolicy-M can be viewed as a simplified attempt to model this application.
> - (3) Visually grounded guidelines: manuals for an app or product where bounding boxes are used to localize buttons, since describing such spatial coordinates purely in text is highly challenging for models to interpret.
>
> We believe that although multimodal prompts and multimodal memory are still rare in existing conversational agents, they will become increasingly prevalent in the near future.
>
> # Q1: Pointers to the estimation of policy prompt lengths
> Yes, we acknowledge that the range is quite large. As also footnoted in the paper, the main reason is that the system prompt (lengths) for mainstream conversational agents such as GPT and Claude are business secrets and are not publicly available. Our estimation is mostly based on online sources. Here are some references:
> - https://www.oreilly.com/radar/unpacking-claudes-system-prompt/?utm_source=chatgpt.com
> - https://www.reddit.com/r/PromptEngineering/comments/1mknun8/i_have_extracted_the_gpt5_system_prompt/
> - https://saharaai.com/blog/writing-ai-system-prompts?utm_source=chatgpt.com

---

> > ### Author Response · Authors · 2025-11-21
> >
> > # Q3: Distinction with deliberative alignment (Guan et al., 2024)
> > Here is an in-depth breakdown of the distinctions between this work and deliberative alignment:
> >
> > - **Task and Domain:**
> > Deliberative alignment focuses solely on safety-related domains, such as safety guidelines and jailbreak prevention, whereas we aim at more general instruction-following (ClevrPolicy) and tool-using (GTAPolicy) scenarios.
> >
> > - **Modality:**
> > As the reviewer already mentioned, our work targets multimodal models and involves multimodal inputs and policy content, while deliberative alignment is restricted to text-only settings.
> >
> > - **Algorithm:**
> > Deliberative alignment adopts a two-stage training framework consisting of SFT and RL, where the RL stage additionally requires a reward model. Its code is not open-sourced, and the description of the RL algorithm is limited; we suspect it uses a PPO-based approach.
> > In contrast, our proposed TriMPI is a three-stage training framework consisting of CPT, SFT, and RL. Our RL stage follows the RLVR framework and further introduces PolicyRollout, a new group-based policy optimization algorithm that is generalizable to any policy-internalization setting.
> >
> >
> > # Q4: Writing improvements
> > We thank the reviewer for the suggestions on improving the writing and have incorporate these changes in the revised manuscript. A quick clarification regarding Section 4: we did not add another paragraph for the CoT SFT stage because it is identical to the baseline section above, so we direct readers there due to space limits.

---

### Official Review · Reviewer_kmvm · 2025-11-01

**Soundness:** 3
**Presentation:** 3
**Contribution:** 3
**Rating:** 8
**Confidence:** 4

**Summary:**

The paper introduces Multimodal Policy Internalization (MPI), a task and methods for training multimodal conversational agents to follow complex, reasoning‑intensive policies without carrying those policies in the inference prompt. The authors propose TriMPI, a three‑stage training recipe: (1) Visually‑Masked Continual Pretraining (VM‑CPT) to inject policy knowledge by language modeling the combined (policy + task) streams while masking visual tokens; (2) CoT‑SFT, which teaches explicit policy‑guided reasoning on task data; and (3) RL fine‑tuning with a new rollout augmentation called PolicyRollout (PoRo) that adds policy‑aware trajectories during exploration but keeps training/inference aligned by applying policy gradients only to the no‑policy path. They also release two benchmarks: ClevrPolicy (synthetic, controllable policy complexity; a text‑only variant and an image‑augmented policy variant) and GTAPolicy (real‑world images and queries for tool‑use with versioned, user‑conditional rules; low‑data regime). TriMPI yields large gains over SFT and in‑context baselines, maintains general capabilities, and substantially reduces prompt tokens and prefil time once policies are internalized.

**Strengths:**

The main strength of the paper is as following:

1. The authors address a practical, under‑explored problem: handling long, reasoning‑intensive policies for decoder‑only models. The motivation is clear and quantified.
2. Two new datasets are introduced. ClevrPolicy (synthetic, controllable complexity; text‑only and image‑augmented policy variants) and GTAPolicy (real images and queries; tool metadata and versioned rules in a low‑data regime) provide controlled benchmarks.
3. The three components (VM‑CPT, CoT‑SFT, RL) are well motivated and ablated; PolicyRollout improves over GRPO/DAPO by enabling policy‑grounded exploration. And we can clearly observed that SFT part is not enough to train the model to compress the long prompts.
4.  The paper evaluates policy updates, policy knowledge referral, catastrophic forgetting, and efficiency, which is crucial for this method, because, generally, on-inference prompt extension with the policy description seems more robust approach in comparison to the specific training.

**Weaknesses:**

Despite the strong points of the research, I noticed a few weaknesses

1. Risk of overfitting and external validity. While TriMPI maintains general abilities (MMMU‑Pro/MMLU‑Pro) and handles policy updates (Policy Override), broader real‑policy evaluations (e.g., long stylistic/safety guidelines) would further validate external generalization.
2. The three‑stage pipeline, particularly RL, increases implementation and tuning burden; DAPO vs. GRPO behavior on small data underscores this.
3. Although the authors argue soft prompts are task‑specific and hurt robustness, a direct empirical comparison to strong prompt‑tuning/gist‑token baselines would strengthen the case.

**Questions:**

My questions to the authors are the following:

1. I could miss something, but you notice that you apply vision masking in the first training stage, isn't it a common practice for almost all adapter-based multimodal training, to mask vision input from calculating cross-entropy loss? Or there is some specific trick applied in your research. Could you, please, provide more detailed explanation for this stage?
2. Beyond Policy Override, do you evaluate OOD generalization across visual domains or unseen tool types?
3. How do failure modes break down (policy‑branching vs. perception vs. tool‑argument formatting), especially on GTAPolicy?
4. What are the most common overfitting patterns you observed during training, and can VM‑CPT or PoRo be adapted to mitigate them further?

---

> ### Author Response · Authors · 2025-11-21
>
> **We thank reviewer kmvm for the detailed and insightful comments. We appreciate the reviewer’s acknowledgement of our strengths and for contributing to a positive and professional discussion. We address the remaining questions and suggestions below, and we have reflected these changes in the updated manuscript.**
>
> # W1: Broader real-policy evaluations for validating external generalization
> We agree with the reviewer that it would be ideal to include additional benchmarks containing “A long policy + N <question, answer> pairs” to further demonstrate the preservation of general policy-following capabilities. However, to our knowledge, there are very few (if any) existing benchmarks that satisfy this setting, especially in the multimodal domain. This limitation is also a primary motivation behind our construction of the two new benchmarks.
>
> Nevertheless, we include an **additional benchmark, WildGuardTest [1]**, to further evaluate the model’s safety-related behavior as suggested. Given a prompt and a model response, the task is to determine whether the response is harmful (e.g., involving privacy violations, misinformation, harmful language, or malicious uses). The following table reports the accuracy (%) on WildGuardTest for models after training with ClevrPolicy-T, ClevrPolicy-M, and GTAPolicy. We show that the proposed TriMPI method maintains the most consistent performance across different policy-internalization settings, preserving most of the overall capabilities. In contrast, the SFT and GRPO baselines exhibit larger variance and, in particular, suffer a substantial performance drop under the ClevrPolicy-M setting.
>
>
>
> | Model | ClevrPolicy-T -> WildGuard | ClevrPolicy-M -> WildGuard | GTAPolicy -> WildGuard | AVG
> |-|-|-|-|-|
> | Base (Qwen2.5-VL-7B)| 87.24 | 87.24 | 87.24 | 87.24
> | SFT | 82.04 | 52.95 | 69.22 | 68.07
> | SFT + GRPO | 83.21 | 47.69 | 69.87 | 66.92
> | **Ours (TriMPI w/ PoRo-GRPO)** | 74.90 | 76.71 | 76.77 | 76.13
>
>
> ### Reference for W1:
> [1]  Han, Seungju, et al. "Wildguard: Open one-stop moderation tools for safety risks, jailbreaks, and refusals of llms."
>
>
>
> # W2: Implementation and tuning burden for the three-stage pipeline
> We acknowledge that a three-stage pipeline requires additional compute and implementation effort. We have tried our best to keep the methods simple yet effective, and the overall pipeline easy to automate. For example, the implementation of PolicyRollout is reasonably straightforward, where the core modifications are under 60 lines of code. We will also make the code and data publicly available for research purposes in the near future. We believe that the resulting improvements in both performance and efficiency are worth the extra work.
>
> # W3: Comparison with soft prompt tuning
> We thank the reviewer for the suggestion to compare with soft-prompt tuning, and we agree that such an analysis would provide additional insights. We did not include this comparison in the current work for several reasons:
> - (1) the task settings are misaligned: because soft-prompt tuning introduces additional parameters, we find it difficult to justify a strictly apples-to-apples comparison.
> - (2) in terms of utility: recent literature has shifted away from soft-prompt or prompt-tuning-based approaches toward alignment-based methods (e.g., deliberative alignment). Soft prompts are inherently task-specific, requiring different embeddings to be inserted or removed depending on the task, which presupposes advance knowledge of the task and limits their applicability in large, general-purpose foundation models.
> Moreover, because TriMPI is a general training framework, it can in principle be used to train soft prompts as well.
>
> Finally, we found no existing codebase or framework that supports soft-prompt tuning for Qwen-2.5-VL, and integrating it into our SFT (LlamaFactory) and RL (EasyR1) frameworks would require non-trivial engineering efforts. Due to the limited rebuttal time, we have to leave this comparison to future work.

---

> > ### Author Response · Authors · 2025-11-21
> >
> > # Q1: Additional explanation on the VM-CPT stage
> > Yes, your understanding is correct, we apply the mask to avoid computing the cross-entropy loss on continuous visual tokens. We use VM-CPT instead of CPT because, in the traditional CPT setting, the cross-entropy loss is computed on ALL tokens; adding the “visually masked” term makes this distinction explicit.
> > This question also echoes one of our discussions in the limitations section: we believe it is a promising future direction to explore more sophisticated continual pretraining strategies beyond simply masking out the visual tokens, in order to further improve the model’s ability to understand multimodal content within policies.
> >
> > # Q2: Evaluate on OOD visual domains and unseen tools beyond policy override
> > This is partly supported by the experiments on MMMU-Pro, where the visual domain and tasks differ entirely from those seen during training. However, due to the scarcity of multimodal policy datasets (as also discussed in W1), we are unfortunately unable to conduct additional OOD evaluations. We do agree that this would further strengthen the experimental section. Ideally, with sufficient policies and data, we would set aside certain attributes (e.g., a particular color) or tools (e.g., object detection) exclusively for evaluation.
> >
> >
> > # Q3: Failure modes breakdown
> >
> > We provide an additional analysis on failure mode breakdown for ClevrPolicy and GTAPolicy. We have included the **additional qualitative examples in the updated manuscript in Figures 11, 12 and 13 (in Appendix).**
> >
> > **ClevrPolicy:** On ClevrPolicy, we find that both perception errors and reasoning errors occur:
> > - Perception Errors
> >     - Perception error on existence: In clustered scenes, the model may fail to detect the existence of some occluded objects.
> >     - Perception error on attributes: This usually happens when multiple objects with shared attributes are present. For instance, two grey cylinders, one large metal and one small rubber, where the larger one is more visible and the smaller one is more hidden. The model may incorrectly perceive that the size of the rubber grey cylinder is large.
> > - Reasoning Errors
> >     - Branching error: This is one of the most common reasoning errors, where the model branches to a non-existent Condition for the current policy. We include further discussion in Q4.
> >     - Knowledge error: The model may correctly branch to an existing Condition, but its referral to the content of that Condition is hallucinated.
> >
> > **GTAPolicy:** Most errors on GTAPolicy are reasoning errors, since the perception of the images is less challenging in this dataset. One typical reasoning error involves incorrectly incorporating the interaction history when deciding which tool to use for the current step. For example, consider the following query: “What is the mood of the white eggs in the picture?”
> > The image shows several cartoon egg characters with different facial expressions. Given a previous action of calling the “TextToBbox” tool, which returns the bounding box of the region of interest, the model should next call “RegionAttributeDescription_v2” to obtain a description of that specific region. However, the model incorrectly calls “ImageDescription_v3”, which is intended for describing the entire image.
> >
> > # Q4: What is the most common overfitting pattern?
> > We observe a typical overfitting pattern is generating referral to Conditions/Rules that does not exist in the corresponding policy (the branching error).
> >
> > We conducted an **additional analysis** to see whether the proposed method can mitigate this failure mode. Particularly, we compare the distribution of referenced Condition IDs in model-generated responses against those in a synthesized Gold CoT response for the ClevrPolicy dataset. We evaluated SFT, SFT + GRPO, and TriMPI models, and the corresponding plots are provided in **Figure 14 (in Appendix) in the updated manuscript**.
> >
> > Our results show that GRPO does not meaningfully reduce the discrepancy in Condition ID distribution relative to the SFT baseline (Average Absolute Difference: 38.72 vs. 39.74), indicating that GRPO alone does not sufficiently improve referral accuracy to the policy. In contrast, TriMPI achieves a substantial reduction in this discrepancy (Average Absolute Difference: 7.07), demonstrating that TriMPI enables much more grounded and policy-aligned reasoning.

---

### Author Response · Authors · 2025-12-03
**Summary of Rebuttal Discussion**

Dear Area Chair,

Thank you for your time and effort in reviewing our submission. Following the recommendation of the ICLR 2026 Program Chairs, we would like to provide a brief factual summary of the author-reviewer discussion. **All interactions happen prior to the incident. Our goal is to accurately reflect the reviewers’ active engagement during the process, which we greatly appreciate.**

- **All reviewers** provided detailed and professional comments, which we carefully considered. We conducted additional experiments and analyses. The major updates are summarized as follows:
  - Additional baseline with policy-in-context inference
  - Additional evaluation on WildGuardTest
  - Additional qualitative analysis on failure modes and overfitting patterns
  - Additional quantitative analysis on insufficient grounding of policy
  - Additional discussion on vision-specific aspects
  - Writing improvements for clearer references to Appendix content

- **Reviewer 2FLG** responded to our initial rebuttal on Nov 28 and asked follow-up questions. We then provided further explanation of the observation and acknowledged that it represents an interesting open question. We did not have the opportunity to receive a response from the reviewer before the discussion period was frozen.

We sincerely thank all reviewers for their tremendous efforts throughout the discussion period, which truly helped us strengthen the submission.

---

### Meta-Review · Area_Chair_txvX · 2025-12-05

**Summary:**

The reviewers consistently praised the paper for proposing a practical and well-motivated task as well as  new benchmarks. However, several concerns about the evaluation.
Reviewer 2FLG initially questioned the lack of vision-specific explorations in the methodology and requested comparisons against baselines where the policy is retained in-context during inference to understand the trade-offs.
Reviewer DXvn raised concerns about the complexity of the multi-stage training pipeline compared to prompt-based methods and the reliance on self-constructed datasets rather than real-world data.
 Reviewer kmvm focused on the risks of overfitting, the lack of OOD generalization evaluation, and the absence of comparisons to soft-prompting techniques.

**Reviewer Concerns:**

The rebuttal addresses most of the reviewers' concerns through additional analysis and experiments.

 Specifically, the authors addressed Reviewer 2FLG's request for an in-context inference baseline. While the result was counterintuitive (internalized models performed better without the context than with it), the authors provided a plausible explanation regarding training distribution mismatch.

The authors also addressed Reviewer kmvm's concerns about external validity and overfitting by including results on the WildGuardTest benchmark and providing a detailed analysis of failure modes.

 Reviewer DXvn's question regarding the necessity of multimodal policies was addressed with concrete examples of visual constraints in real-world scenarios.

The concern from Reviewer DXvn is the reliance on synthetic/constructed datasets (ClevrPolicy and GTAPolicy).

**Reviewer Scores:**

Reviewer DXvn and Reviewer kmvm both initially gave high scores of 8 (Accept). Given that the authors provided the requested failure mode analysis, additional safety benchmarks (WildGuardTest), and clarified the distinction from related work, these reviewers would likely have maintained their high scores or strengthened their confidence in the acceptance.

 Reviewer 2FLG (Score: 6) questioning the counterintuitive results of the new baseline. If the discussion continued, Reviewer 2FLG likely would have maintained their score of 6 or potentially raised it to a 7, as the authors demonstrated effort to run the requested baseline and offered an explanation for the results, thereby satisfying the reviewer's request.

The problem of over-reliance on synthetic/constructed datasets is challenging to address.  Given that reviewer DXvn initially gave a high score of 8, it will not be a reason to lower the score.

---

### Decision · Program_Chairs · 2026-01-26

Accept (Poster)